# Self-Supervised Learning of Graph Representations for Network Intrusion Detection

**Lorenzo Guerra**[1,2]    **Thomas Chapuis**[2]    **Guillaume Duc**[1]

**Pavlo Mozharovskyi**[1]    **Van-Tam Nguyen**[1]

[1]LTCI, Télécom Paris, Institut Polytechnique de Paris, Palaiseau, France
`{name.surname}@telecom-paris.fr`

[2]Ampere Software Technology, Guyancourt, France
`{name.surname}@ampere.cars`

## Abstract

Detecting intrusions in network traffic is a challenging task, particularly under limited supervision and constantly evolving attack patterns. While recent works have leveraged graph neural networks for network intrusion detection, they often decouple representation learning from anomaly detection, limiting the utility of the embeddings for identifying attacks. We propose GraphIDS, a self-supervised intrusion detection model that unifies these two stages by learning local graph representations of normal communication patterns through a masked autoencoder. An inductive graph neural network embeds each flow with its local topological context to capture typical network behavior, while a Transformer-based encoder-decoder reconstructs these embeddings, implicitly learning global co-occurrence patterns via self-attention without requiring explicit positional information. During inference, flows with unusually high reconstruction errors are flagged as potential intrusions. This end-to-end framework ensures that embeddings are directly optimized for the downstream task, facilitating the recognition of malicious traffic. On diverse NetFlow benchmarks, GraphIDS achieves up to 99.98% PR-AUC and 99.61% macro F1-score, outperforming baselines by 5–25 percentage points.[1]

## 1  Introduction

As more devices are connected to the internet, the frequency and sophistication of cyberattacks continue to rise, exposing vulnerabilities across diverse network environments, from enterprise infrastructures to embedded devices. These threats are difficult to anticipate, as they evolve rapidly and exploit previously unknown weaknesses, making timely and accurate detection essential for protecting critical services. Although numerous network intrusion detection methods have been proposed, most rely on supervised learning and depend on large volumes of labeled data, which are expensive and labor-intensive to obtain. Additionally, supervised models are typically trained on known patterns, limiting their ability to detect novel threats and requiring frequent retraining to remain effective. Unsupervised methods have also been explored, but they often struggle to capture the complexity of network traffic, reducing their effectiveness against subtle or sophisticated attack patterns [1].

Self-supervised learning has emerged as a promising alternative by enabling the extraction of rich representations from data without the need for explicit labels. In the context of network intrusion detection, recent studies have leveraged network topology through graph-based methods [2], achieving substantial improvements over traditional techniques. As computer networks can be naturally

---

[1]Code and pre-trained models: `https://github.com/lorenzo9uerra/GraphIDS`.

39th Conference on Neural Information Processing Systems (NeurIPS 2025).

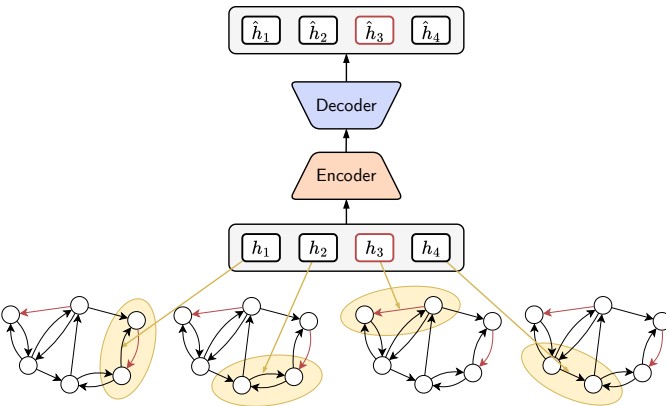

Figure 1: Overview of GraphIDS: the model detects network intrusions by evaluating the reconstruction error of graph-based flow embeddings. Flows representing attacks (highlighted in red) typically yield higher reconstruction errors, as they deviate from normal communication patterns.

represented as graphs, where nodes represent hosts and edges represent communication flows, we can employ Graph Neural Networks (GNNs) to learn meaningful representations that facilitate modeling the normal network behavior. However, unlike typical GNN applications, where the relevant information is encoded in node features and their connections, network intrusion detection primarily encodes critical information within the edges, which reflect either normal host interactions or malicious activities.

Nevertheless, most existing approaches decouple graph representation learning from the anomaly detection task, which limits their ability to learn meaningful embeddings that are directly useful for detecting intrusions. Moreover, the self-supervised pretext tasks frequently rely on the availability of negative samples or prior knowledge to construct them—assumptions that often do not hold in practical intrusion detection scenarios, where labeled anomalies or well-defined negative examples are scarce or entirely unavailable.

To address these limitations, we propose GraphIDS, a self-supervised model jointly trained to reconstruct local graph representations of benign network traffic, using reconstruction error to identify potential intrusions (Figure 1). In network intrusion detection, anomalous behavior often manifests through irregular communication patterns and deviations from typical structural or statistical relationships. The GNN encoder captures local topological context by modeling the immediate neighborhood of each flow, integrating both structural and edge-level features that characterize the underlying structure of benign traffic.

The autoencoding objective, applied to batches of flow embeddings, enables the Transformer to learn broader contextual dependencies and co-occurrence patterns across the network. During training, we apply attention masking as a structured regularizer, randomly disabling a subset of attention links while keeping all input embeddings visible. This encourages the model to infer from partial context and learn a stable distribution of normal behavior. At inference time, flows that significantly deviate from this learned distribution exhibit high reconstruction errors and are flagged as potential intrusions. This end-to-end framework enables the model to generalize effectively to unseen attack types and dynamic network environments without relying on supervision or negative samples.

We evaluate our model on multiple NetFlow-based datasets for Network Intrusion Detection Systems (NIDS), including NF-UNSW-NB15 and NF-CSE-CIC-IDS2018. We use both their second version (v2), which includes 43 NetFlow features, and their third version (v3), which extends these with 10 additional temporal features, resulting in a total of 53 features [3][4].

This work makes the following key contributions:

1. We present GraphIDS, a novel self-supervised framework that jointly trains a GNN encoder and a Transformer-based masked autoencoder to reconstruct local representations of benign network traffic. By leveraging both topological and contextual information in an end-to-end manner, GraphIDS directly optimizes flow embeddings for anomaly detection, without

relying on labeled data or prior knowledge of attack patterns. To the best of our knowledge, this is the first application of a jointly trained GNN-Transformer architecture for network intrusion detection (Section 3).

2. We conduct extensive experiments on NIDS datasets, covering diverse network environments and a wide range of attack types. GraphIDS achieves state-of-the-art performance, outperforming existing methods by 5% to over 25% in both macro F1 and PR-AUC. These results demonstrate its strong generalization to previously unseen attacks, highlighting its practical effectiveness for real-world intrusion detection (Section 4).

## 2 Related Work

**Self-supervised Masked Modeling:** Masked modeling has emerged as a fundamental technique for self-supervised learning across various data modalities. In natural language processing, BERT is trained to reconstruct randomly masked tokens through a bidirectional Transformer, thereby learning to generate word embeddings that are transferable to a wide range of downstream tasks [5]. Similarly, in computer vision, masked autoencoders are trained to reconstruct randomly masked patches of an image by employing an asymmetric encoder-decoder architecture, compelling the model to extract rich contextual representations from the input data [6]. GraphMAE [7] extended this concept to the graph domain by training a graph masked autoencoder to reconstruct masked node features in order to learn meaningful graph representations.

All of these models share a common objective: learning the normal structure or contextual dependencies of the data through a reconstruction-based task. This paradigm is particularly well-suited for anomaly and intrusion detection, as anomalies, such as attacks, typically lead to poor reconstruction, and can thus be effectively identified by quantifying the reconstruction error relative to the original data. For example, Georgescu [8] applied MAE to medical imaging, training the model exclusively on healthy samples to identify abnormal scans based on reconstruction discrepancies. Similarly, BERT-like models have also been adapted to textual anomaly detection, enabling the identification of system malfunctions or malicious activity in log data by capturing deviations from learned normal patterns [9].

**Graph Neural Networks for Supervised Intrusion Detection:** GNNs provide a principled framework for integrating graph topology with node/edge attributes into learned embeddings. Foundational models such as Graph Convolutional Networks (GCN) formulate spectral convolutions on graphs and learn node embeddings by aggregating information from local neighborhoods [10]. GraphSAGE [11] introduces an inductive approach that learns to aggregate and transform features from a node's neighbors to embed unseen nodes or graphs. These architectures propagate node features through edges to encode local topology and features simultaneously. More advanced variants, including Graph Attention and Graph Transformers, enhance expressivity, but the fundamental principle remains the same: summarizing each node's local structure into its embedding.

While standard GNNs primarily focus on node embeddings, edge-level representations can also be learned by various techniques, such as employing the line graph, or combining end-point embeddings. For example, E-GraphSAGE [12] explicitly incorporates edge features into GraphSAGE by performing message passing that aggregates edge attributes and node attributes together. With network flow data represented as a graph of endpoints connected by edges (flows), E-GraphSAGE becomes crucial for capturing edge features (e.g., flow statistics) along with network topology. Alternatively, Friji et al. [13] propose a novel graph construction in which each network flow is represented as a node within a directed "flow graph". This formulation allows complex behaviors such as multi-step attacks and spoofing to emerge as recognizable graph patterns. In another example, Zhou et al. [14] introduces a NIDS that exclusively leverages topology information to detect botnet activity, such as the hierarchical organization of centralized botnets or the fast-mixing structure of decentralized ones. Collectively, these studies highlight the inherent capability of GNNs to model rich structural properties in network data. However, they rely on labeled traffic, which constrains their deployment in practical, large-scale environments where annotated data is scarce or unavailable.

**Graph Neural Networks for Self-Supervised Intrusion Detection:** To address the limitations imposed by the need for labeled data, Caville et al. [2] recently introduced Anomal-E, a self-supervised framework designed to detect attack patterns without relying on flow labels. The approach begins

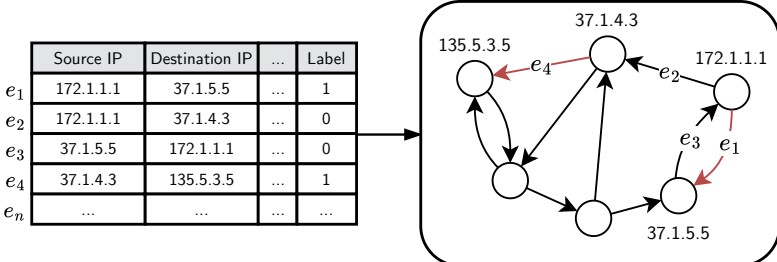

Figure 2: Illustration of the graph construction process. Network flows are transformed into a directed graph, where nodes represent hosts (IP addresses) and edges correspond to communication flows between them. Edge features capture flow statistics such as packet count, byte count, and protocol information.

with the pretraining of a GNN-based encoder using a contrastive pretext task, which is then used to generate representative flow embeddings. Afterwards, a separate unsupervised algorithm is applied to identify the anomalous embeddings.

In contrast to Anomal-E, our method adopts an end-to-end training paradigm that jointly optimizes a GNN with the Transformer encoder-decoder within a masked reconstruction framework. This design enables the GNN to learn representations that are intrinsically aligned with the intrusion detection objective. Moreover, our approach eliminates its reliance on contrastive learning and negative samples, thereby removing the necessity for prior knowledge of attack patterns. Instead, the model is trained exclusively on benign traffic, which is generally more accessible and easier to obtain in real-world environments. This design choice allows the system to robustly capture the normal behavior of network traffic, making it particularly suitable for practical deployment in dynamic and evolving threat landscapes, where the nature of attacks can change significantly over time.

## 3 Method

### 3.1 Graph Construction and Framework Introduction

We define a network flow as a unidirectional sequence of packets that share at least five attributes [15]: ingress interface, source IP address, destination IP address, IP protocol number, and IP Type of Service. When UDP or TCP protocols are used, source and destination ports are also included, providing additional granularity to uniquely identify individual flows.

Using the collected network flows, we construct a graph representation of the computer network, where nodes correspond to hosts (identified by IP addresses) and edges represent communication flows. Each flow defines a directed edge from a source node (source IP address) to a destination node (destination IP address), with edge features derived from flow statistics that provide a high-level summary of the communication. The overall graph construction process is illustrated in Figure 2.

Assuming the ability to monitor communications among network hosts to be protected from malicious activity, we begin by collecting and aggregating network flows to obtain a comprehensive view of network activity over time. These flows are then used to construct the graph, on which we apply a GNN to compute flow-level embeddings that include neighborhood context, integrating information about recent local communications. To learn the normal behavior of the network, we train the model to reconstruct these local graph representations, enabling it to detect anomalies based on reconstruction errors.

### 3.2 Graph Representation Learning

To incorporate local graph context into flow embeddings, we employ E-GraphSAGE [12], an extension of the original GraphSAGE [11] which includes edge features during the embedding process. As edges are the most informative elements in our graphs, this approach allows GraphIDS to integrate local topological patterns into each flow representation, improving its ability to distinguish between benign and malicious activity.

Unlike early GNN models such as the original GCN [10], which are mainly transductive (i.e., they can only make predictions on nodes seen during training), E-GraphSAGE is designed for inductive learning. It can generalize to unseen edges and graphs at inference time by learning aggregation functions over neighborhoods rather than depending on a fixed adjacency matrix.

To ensure scalability, we use neighborhood sampling, which makes the method efficient in both memory and computation and enables training on large graphs using mini-batches. In addition, by not relying on a fixed graph structure, E-GraphSAGE is well-suited for network intrusion detection, where the underlying graph is dynamic and affected by noise.

Although in principle deeper GNN architectures could be employed, we restrict our GNN to a 1-hop neighborhood to efficiently capture informative local context, a design choice validated by our ablation study in Appendix C.5. In network intrusion detection, the immediate neighborhood of a flow often reveals meaningful patterns that are indicative of malicious behavior, such as unusual connections or bursty communications. Global co-occurrence patterns, which may span multiple flows or hosts, are instead captured by the Transformer operating over batches of flow embeddings. This separation of roles allows GraphIDS to jointly model both local and global dependencies without incurring the computational overhead of deeper GNN layers.

To further reduce memory usage when processing graphs with a very large number of edges relative to the number of nodes, we impose a configurable limit on each node's fanout during sampling. This fanout is treated as a tunable hyperparameter.

### 3.3 Masked Autoencoder

To learn global co-occurrence patterns, our model employs a Transformer-based masked autoencoder designed to reconstruct benign graph representations. This component operates on batches of flow embeddings, $\mathbf{H} = \{h_1, \ldots, h_n\}$, produced by the GNN layer, where each $h_i \in \mathbb{R}^{d_{\mathrm{gnn}}}$. We construct each batch by concatenating 64 flow windows of 512 flows each, resulting in a sequence of $n = 32,768$ embeddings. Before being passed to the Transformer, these embeddings are projected to a lower-dimensional space $\mathbb{R}^{d_{\mathrm{model}}}$ through a linear layer, encouraging the model to learn more compact representations and reducing computational cost.

The Transformer consists of an encoder and a decoder, each built from a stack of identical blocks. Each block contains a multi-head self-attention module followed by a position-wise feed-forward network. Both sub-layers are wrapped with residual connections and layer normalization. During training, we apply a symmetric binary attention mask that disables attention between a randomly sampled subset of positions. Unlike conventional masked autoencoders that remove input tokens, our approach applies a mask directly to the attention mechanism, functioning as a form of structured dropout on the attention weights to improve generalization. This process encourages the model to build more robust and distributed representations by preventing over-reliance on specific contextual links. Random masking is disabled at inference.

The encoder processes the projected embeddings to produce contextual representations. The decoder then takes the same projected sequence as reconstruction queries and attends to the encoder outputs via cross-attention to compute per-position reconstructions. Finally, an output projection layer maps these reconstructed embeddings, $\hat{\mathbf{H}}'$, back to the original GNN embedding space to produce the final output, $\hat{\mathbf{H}} = \{\hat{h}_1, \ldots, \hat{h}_n\}$. The anomaly score for each flow is its squared reconstruction error, computed in the original GNN embedding space:

$$s_i = \|h_i - \hat{h}_i\|^2 \tag{1}$$

During inference, higher anomaly scores identify potential attacks.

Let $n_{\mathrm{valid}}$ be the number of non-padded embeddings in a batch. The model is trained end-to-end by minimizing the Mean Squared Error (MSE) over these embeddings:

$$\mathcal{L}_{\mathrm{MSE}} = \frac{1}{n_{\mathrm{valid}}} \sum_{i=1}^{n_{\mathrm{valid}}} s_i = \frac{1}{n_{\mathrm{valid}}} \sum_{i=1}^{n_{\mathrm{valid}}} \|h_i - \hat{h}_i\|^2 \tag{2}$$

The gradient of the MSE loss is backpropagated through both the Transformer and the GNN, jointly updating their parameters. This aligns the training objectives of the two components, allowing the model to capture both local structural context and global co-occurrence patterns without requiring labeled attack data.

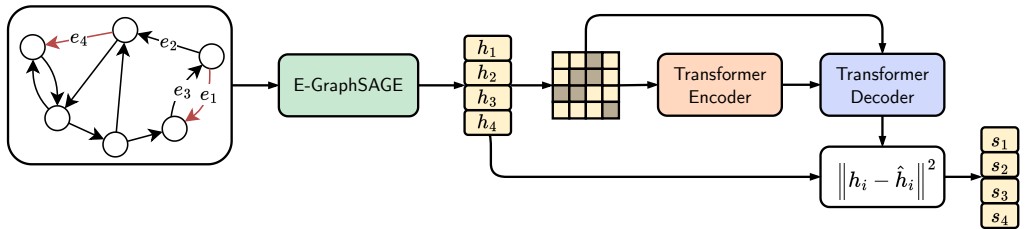

Figure 3: Overview of the full training pipeline. During inference, attention masking is omitted. The reconstruction errors $s_1, s_2, \ldots, s_n$ serve as anomaly scores for each network flow.

## 3.4 Summary of the Pipeline

The full training pipeline is illustrated in Figure 3. Network flows are first collected and processed into a graph, where edges represent communications between hosts. Using neighborhood sampling, E-GraphSAGE integrates local context into edge embeddings $h_1, h_2, \ldots, h_n$. These embeddings are then batched, projected, and passed to the masked autoencoder. The model's parameters are optimized end-to-end by backpropagating the gradient of the MSE loss. This gradient flows from the output of the decoder back through both the Transformer and the GNN, allowing for the joint update of all model parameters. This ensures that the GNN learns to produce representations that are not only structurally informative but also optimized for the reconstruction task performed by the Transformer.

At inference time, the model processes batches of flow embeddings without masking. Attacks are detected by applying a threshold to the reconstruction error, calculated as the squared L2 norm between the original GNN embeddings and their final reconstructions. Flows with higher reconstruction errors are flagged as potential intrusions, as they deviate from the learned patterns of normal network traffic.

# 4 Experimental Evaluation

## 4.1 Datasets

For our experiments, we selected two widely used datasets for evaluating NIDS: UNSW-NB15 [16], captured in a smaller-scale network environment, and CSE-CIC-IDS2018 [17], which captures traffic from a significantly larger infrastructure. Specifically, we used their NetFlow-based versions, which correct flow extraction errors introduced by CICFlowMeter [18] and provide a standard feature set including IP addresses, allowing us to construct the graphs and fairly evaluate the models across datasets. We conduct our experiments both on their second version [4], which includes 43 NetFlow features, and their recently released third version [3], which adds 10 additional temporal features[2].

Table 1 summarizes the main characteristics of the datasets. By selecting two datasets collected from different network environments and with different attack scenarios, we can verify if our approach is able to generalize across different settings. The datasets feature extensive encrypted traffic (e.g., HTTPS, SSH) and diverse communication patterns, representing realistic, modern network environments. We note that the differences in statistics between the second and third versions of each dataset are due to variations in the NetFlow aggregation process during feature extraction, although the details are not documented by the original authors.

## 4.2 Data Preprocessing

Before training, we replace missing or invalid values with zeros and split the dataset into 80% for the training set, 10% for the validation set, and 10% for the test set, preserving the class distribution of attack types. We normalize all NetFlow features to the interval $[0, 1]$ using Min-Max scaling. The scaler is fitted on the training set, which contains only benign flows, and then applied to transform

---

[2]The datasets are available under the "Permitted reuse with commercial use restriction" license (`https://guides.library.uq.edu.au/deposit-your-data/license-reuse-noncommercial`).

Table 1: Statistics of the NetFlow-based datasets, varying in scale and anomaly prevalence.

| Dataset | Number of Flows | Number of Hosts | Anomaly Ratio |
|---|---|---|---|
| NF-UNSW-NB15-v2 | 2,390,275 | 44 | 3.98% |
| NF-UNSW-NB15-v3 | 2,365,424 | 44 | 5.40% |
| NF-CSE-CIC-IDS2018-v2 | 18,893,708 | 255,042 | 11.95% |
| NF-CSE-CIC-IDS2018-v3 | 20,115,529 | 205,801 | 12.93% |

both the validation and test sets. To avoid extreme input values, any scaled value falling outside the original training range is clipped to the $[-10, 10]$ range. For v3 datasets, timestamps are discarded, as our experiments showed that these features introduced unnecessary noise, with no performance improvement (Appendix C).

The graph is then constructed, as specified in Section 3.1, by using IP addresses to identify hosts, while the remaining flow features are assigned to each edge. Since attack behaviors are directional, we use a directed graph to preserve this information, allowing us to easily correlate the classification results with the network flows. We also note that one of the baseline models considered, Anomal-E, based on its original implementation and specification, is unable to efficiently process the full graph of NF-CSE-CIC-IDS2018 on our hardware. Therefore, in this case, we downsample the dataset to 20% of its original size, maintaining the proportions of the attack types.

### 4.3 Training and Anomaly Detection Procedure

**Mini-Batch Strategies.** After preprocessing, training proceeds with different mini-batch strategies for each component of our architecture. For E-GraphSAGE, we employ neighborhood sampling to manage the computational demands of graph processing. First, we divide the full set of edges into mini-batches (with batch sizes ranging from 16,384 to 32,768 depending on the dataset). For each edge in a mini-batch, we sample edges from its 1-hop neighborhood up to a dataset-specific fan-out limit.

For the Transformer encoder-decoder, we process the graph embeddings in fixed-size batches of 64, where each batch contains 512 graph-based flow representations. The separate batching mechanisms reflect the fundamentally different ways the GNN and Transformer components process information: neighborhood-based for the former and context-based for the latter.

Since our approach computes embeddings using batched processing and randomly sampled neighboring edges, it does not require a fixed adjacency matrix. This makes its application practical for evolving network environments, where connections to new hosts are constantly established. In realistic deployments, memory usage can be controlled by applying a graph-level time window that discards outdated edges when new ones are added.

**Implementation and Optimization Details.** We train GraphIDS for a maximum of 100 epochs using the AdamW optimizer on a machine with an NVIDIA A100 GPU, 32 GB of RAM, and 8 CPU cores of an AMD EPYC 7302 processor. We implemented our model using PyTorch 2.3.1 and DGL 2.3.0. Random seeds were fixed to ensure reproducibility. Table 2 reports the training time for each dataset, which varies depending on early stopping. Across all experiments, including hyperparameter optimization and baselines, the total compute time was roughly 976 GPU hours.

As this architecture requires careful tuning, we thoroughly optimize several hyperparameters for each dataset, including learning rate, weight decay coefficients, edge embedding dimension, autoencoder latent dimension, number of transformer layers, dropout rate and the parameters related to the batching strategies. We initially explored coarse configurations through grid search, followed by Bayesian optimization to tune the hyperparameters based on validation Precision-Recall Area Under Curve (PR-AUC).

Our experiments showed that the model reaches optimal performance when stronger regularization is applied to the GNN encoder compared to the masked autoencoder. The GNN encoder typically benefits from higher weight decay coefficients (up to 0.6) and dropout rates (up to 0.7), while the masked autoencoder performs better with more moderate regularization (weight decay from 0.01 to 0.05 and dropout rate from 0 to 0.2). We also observe that an attention mask ratio of 0.15 provides a

Table 2: Total number of parameters, training time and inference time of GraphIDS across all datasets. Inference is performed in batches; the reported time refers to the forward pass and excludes any preprocessing.

| Dataset | # Params | Training Time (h) | Inference Time (µs/sample) |
|---------|----------|-------------------|----------------------------|
| NF-UNSW-NB15-v2 | 0.867M | 0.39 ± 0.27 | 2.27 ± 0.92 |
| NF-UNSW-NB15-v3 | 0.874M | 0.63 ± 0.32 | 3.04 ± 1.25 |
| NF-CSE-CIC-IDS2018-v2 | 0.570M | 1.00 ± 0.55 | 4.89 ± 0.50 |
| NF-CSE-CIC-IDS2018-v3 | 0.572M | 1.46 ± 0.59 | 5.09 ± 1.09 |

good balance between regularization and reconstruction quality across datasets (see Appendix C.4). All code and configurations used for the experiments have been publicly released to ensure full reproducibility.

**Early Stopping.** We use early stopping based on the PR-AUC achieved on the validation set with a patience of 20 epochs. We selected PR-AUC because it provides a fair, threshold-independent evaluation of the model performance. The model checkpoint achieving the highest validation PR-AUC is saved for final evaluation and deployment. Although this procedure assumes access to a small amount of labeled data for validation and threshold tuning, it avoids the scalability and generalization issues of fully supervised models. In practice, a small amount of labeled data can reasonably be collected during deployment to support this step and it may even provide a more reliable and robust thresholding solution for practical applications. Nevertheless, exploring fully unsupervised thresholding strategies remains an interesting direction for future research.

### 4.4 Baselines

To isolate the contribution of each component in GraphIDS, we evaluate the following targeted ablations:

1. **T-MAE**: A Transformer-based masked autoencoder with attention masking trained directly on raw NetFlow features, without any graph-based representation learning. This ablation isolates the contribution of the GNN by evaluating performance without local topological context.

2. **SimpleAE**: A lightweight fully connected autoencoder that replaces the Transformer with a two-layer MLP encoder and a two-layer MLP decoder (ReLU activations). It is trained end-to-end together with E-GraphSAGE on the same reconstruction objective and with the same batching scheme as GraphIDS, isolating the role of self-attention and sequence modeling.

We also compare GraphIDS against anomaly detection baselines:

- **Anomal-E** [2]: A state-of-the-art method using a pre-trained E-GraphSAGE encoder to generate edge embeddings, followed by anomaly detection on those embeddings. Unlike GraphIDS, it lacks joint training and self-attention, isolating the impact of the Transformer reconstruction.

- **CBLOF** [19]: A clustering-based local outlier factor algorithm, representative of traditional unsupervised detection techniques.

- **SAFE** [20]: A recent self-supervised method that transforms tabular network traffic data into image-like representations for masked autoencoder reconstruction, followed by a lightweight novelty detector.

### 4.5 Evaluation Metrics

To fairly evaluate and compare models on NIDS datasets, we use two complementary metrics that are resistant to high class imbalance: a threshold-dependent macro-averaged F1-score and a threshold-independent PR-AUC. The macro F1-score gives equal weight to both classes, preventing the dominant benign class from overshadowing the minority attack class, and provides a balanced

Table 3: Model performance on NetFlow datasets. Multiple values are bolded when differences are not statistically significant.

| Model | Metric | v3 Datasets (w/ temporal features) | | v2 Datasets (w/o temporal features) | |
|---|---|---|---|---|---|
| | | UNSW-NB15 | CSE-CIC-IDS2018 | UNSW-NB15 | CSE-CIC-IDS2018 |
| *External baselines* | | | | | |
| CBLOF | PR-AUC | 0.3658 ± 0.0634 | 0.2638 ± 0.0263 | 0.2102 ± 0.0157 | 0.7822 ± 0.0198 |
| | Macro F1 | 0.7319 ± 0.0225 | 0.6599 ± 0.0130 | 0.7046 ± 0.0140 | 0.8889 ± 0.0068 |
| SAFE | PR-AUC | 0.8946 ± 0.0279 | 0.6294 ± 0.0923 | 0.2044 ± 0.0267 | 0.5222 ± 0.1896 |
| | Macro F1 | 0.9236 ± 0.0086 | 0.4662 ± 0.0016 | 0.5815 ± 0.0164 | 0.4684 ± 0.0001 |
| Anomal-E | PR-AUC | 0.9032 ± 0.0041 | 0.2555 ± 0.0383 | 0.7489 ± 0.0074 | **0.9287 ± 0.0265** |
| | Macro F1 | 0.9459 ± 0.0009 | 0.6709 ± 0.0394 | **0.9156 ± 0.0217** | **0.9410 ± 0.0161** |
| *Ablations* | | | | | |
| T-MAE | PR-AUC | 0.9914 ± 0.0022 | 0.7398 ± 0.0777 | 0.5995 ± 0.0155 | 0.7375 ± 0.0035 |
| | Macro F1 | **0.9933 ± 0.0017** | 0.4707 ± 0.0049 | 0.8039 ± 0.0447 | 0.8453 ± 0.0317 |
| SimpleAE | PR-AUC | **0.9996 ± 0.0007** | **0.8458 ± 0.0498** | 0.7864 ± 0.0608 | 0.9310 ± 0.0126 |
| | Macro F1 | **0.9838 ± 0.0321** | **0.9223 ± 0.0201** | 0.8680 ± 0.1579 | 0.9450 ± 0.0092 |
| *Ours* | | | | | |
| GraphIDS | PR-AUC | **0.9998 ± 0.0007** | **0.8819 ± 0.0347** | **0.8116 ± 0.0367** | 0.9201 ± 0.0238 |
| | Macro F1 | **0.9961 ± 0.0084** | **0.9447 ± 0.0213** | **0.9264 ± 0.0217** | **0.9431 ± 0.0131** |

view of false positives and false negatives. Unlike ROC-AUC, PR-AUC is well-suited for imbalanced datasets as it focuses on the positive class without being influenced by the large number of true negatives [21]. PR-AUC is computed using scikit-learn's implementation with linear interpolation.

To select the optimal threshold for the final classification, we maximize the macro F1-score on a labeled validation set and then apply the same threshold to the test set. This approach ensures a fair comparison across all models while avoiding threshold optimization bias on the test data.

## 4.6 Results

Table 3 presents averaged metrics across multiple random seeds, along with standard deviations. Since Anomal-E combines several anomaly detectors, we report only its best-performing variant (based on PR-AUC) for each dataset. A complete breakdown of Anomal-E's performance across all detectors is available in Appendix B.2.

On the third version of the NetFlow-based datasets, GraphIDS demonstrates a strong performance, effectively leveraging both local topological context and global co-occurrence patterns to improve detection by identifying complex attack behaviors. In the smaller network environment of NF-UNSW-NB15-v3, our ablated T-MAE model, which relies solely on contextual information, performs comparably to GraphIDS. However, on the larger and more diverse NF-CSE-CIC-IDS2018-v3 dataset, T-MAE shows a substantial drop in performance, highlighting the importance of structural information in large-scale settings.

On the second version of the datasets, GraphIDS achieves comparable results to Anomal-E, while clearly outperforming it on NF-UNSW-NB15-v2 in terms of PR-AUC. Although direct comparisons between dataset versions are not possible due to differing flow aggregation and feature sets, we notice that the v3 configuration made intrusion detection more challenging for NF-CSE-CIC-IDS2018 but improved it for NF-UNSW-NB15. These results suggest that both aggregation parameters and NetFlow feature selection should be carefully tuned before deployment, as they can significantly impact detection performance.

Finally, the SimpleAE ablation confirms that the autoencoding objective itself is a strong driver of performance: it performs competitively across datasets and, in some cases, matches PR-AUC achieved with the Transformer. Nevertheless, GraphIDS typically attains higher macro F1 and more stable results, especially on the larger NF-CSE-CIC-IDS2018-v3, which supports the benefit of self-attention for modeling global co-occurrence patterns among flows.

# 5    Conclusion

We introduced GraphIDS, a novel self-supervised framework that jointly trains a graph neural network and a Transformer-based masked autoencoder to learn normal network behavior and detect complex intrusion patterns. To the best of our knowledge, this is the first end-to-end architecture that combines GNNs and masked autoencoding for network intrusion detection. Experimental results demonstrate that GraphIDS achieves state-of-the-art performance across both small-scale and large-scale scenarios, outperforming existing baselines by 5–25 percentage points. With an average inference time of 3.83 µs per sample, the model is also well-suited for real-time deployment, where rapid response is essential.

Like other unsupervised models, GraphIDS assumes a relatively stable network topology, and its performance may degrade under abrupt behavioral shifts, potentially leading to increased false positives or missed detections. Online learning techniques offer a promising avenue to mitigate this limitation by enabling continuous adaptation without requiring full retraining. Furthermore, similarly to other GNN-based approaches, GraphIDS is less effective in single host monitoring scenarios, where limited topological context constrains its representational capacity. Future work could address this by incorporating multimodal data, such as combining network flows with logs or system calls, to improve the detection of intrusions that leave minimal footprints in network traffic alone. Finally, accurately assessing real-world detection latency requires a holistic evaluation of the entire processing pipeline, including data collection, aggregation, preprocessing, and inference.

Overall, our findings underscore the effectiveness of jointly modeling local topological context and global co-occurrence patterns for network intrusion detection. By unifying GNNs and Transformers under a shared reconstruction objective, GraphIDS captures normal network behavior without relying on labeled data or prior knowledge of attack signatures, offering a practical and scalable solution for real-world deployment.

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

# A   Additional Details on GraphIDS and Baseline Models

**CBLOF [19].**   CBLOF is an unsupervised anomaly detection method which identifies outliers based on a clustering algorithm. In this case we use K-means for clustering and we compute the anomaly scores based on the distance to the closest large cluster. As we do for other models, once the anomaly scores are computed, we search for the best threshold on the validation set and we apply it to the test set for the final classification.

**Anomal-E [2].**   Anomal-E is the first model to apply self-supervised GNNs to network intrusion detection, demonstrating significant performance gains when applying anomaly detection methods to local graph representations compared to raw NetFlow features. The model uses an E-GraphSAGE encoder trained via a modified version of Deep Graph Infomax [22], and generates embeddings that are subsequently processed by traditional unsupervised anomaly detection algorithms, including PCA-based anomaly detection [23], Isolation Forest [24], CBLOF [19], and histogram-based outlier score [25]. For our evaluation, we retained the original GNN encoder configuration, which was already tuned for these datasets, as our attempts at further tuning did not yield performance gains. We did, however, explore and adjust the hyperparameters of the downstream anomaly detection components using the same ranges as in the original work, selecting the best-performing one for each dataset to be included in the main table.

During the preprocessing phase, the original authors report using target encoding for categorical features. Although the specific target used for encoding is not clarified in the paper, their public implementation[3] shows that attack labels are directly used as the target variable—introducing label leakage. This ground-truth information encoded in the input features allows downstream models to learn to identify attacks based on those statistics, undermining the validity of the unsupervised learning setting. To ensure a fair comparison and preserve the integrity of the evaluation, we removed the target encoding step in our implementation.

**SAFE [20].**   SAFE is an anomaly detection framework that processes tabular network traffic data by first applying feature selection to discard irrelevant columns. The remaining features are then mapped into a 2D grid to create image-like embeddings, which a lightweight CNN-based masked autoencoder is trained to reconstruct, learning meaningful representations in the process. For novelty detection, the latent embeddings produced by the encoder are passed to a local outlier factor detector [26] to identify anomalies. In our evaluation, we tuned the hyperparameters of the LOF anomaly detection component. However, a brief exploration of alternative hyperparameters for the MAE module yielded no performance improvements. Given the high computational cost of tuning and the MAE's limited discrimination ability, as observed in the original codebase, we concluded that further tuning would offer marginal gains and not meaningfully affect the conclusions.

In our experiments, we adopt different evaluation metrics, as the original implementation computes the F1-score for the normal class, effectively measuring the model's ability to recognize benign traffic rather than attacks[4]. While this choice may be acceptable for balanced classes, it masks the model's real performance on the highly imbalanced datasets we consider.

**T-MAE.**   T-MAE refers to our Transformer-based masked autoencoder component, similar to the one used in GraphIDS but applied directly to raw NetFlow features. We use the same batching strategy (batch size of 64 with 512 flow embeddings per batch) and tune its learning rate, weight decay, and dropout. We found that a higher learning rate proved to be especially beneficial for the performance on the NF-UNSW-NB15-v3. However, despite this adjustment, T-MAE exhibits slower convergence, resulting in significantly longer training times. On average, it requires 2.21 hours per run, compared to just 0.87 hours for GraphIDS.

**SimpleAE.**   The SimpleAE ablation replaces the Transformer with a fully connected autoencoder consisting of a two-layer MLP encoder and a two-layer MLP decoder with ReLU activations. It is trained end-to-end jointly with E-GraphSAGE on the same reconstruction objective, isolating the

---

[3]`https://github.com/waimorris/Anomal-E/`. Apache License 2.0.

[4]`https://github.com/ElvinLit/SAFE/`. No formal license available. Used with permission from the authors for research purposes only.

architectural benefit on top of our end-to-end reconstruction framework. To ensure a fair comparison, we explored different bottleneck dimensions and hyperparameters.

# B    Extended Results and Comparative Analysis

## B.1    Qualitative Analysis of Detection Behavior

Figure 4 summarizes model performance across datasets through precision-recall curves. These plots illustrate that GraphIDS consistently matches or outperforms the baselines across a variety of settings.

To better understand GraphIDS's behavior on specific attack types, we plot the distribution of anomaly scores (by density) for each dataset, as shown in Figures 5, 6, 7, and 8. To maintain clarity, we present representative examples without error bars. Each plot includes the classification threshold, allowing us to visualize which attack types were correctly detected and which ones were missed. In particular, GraphIDS's lower performance on NF-UNSW-NB15-v2 is due to a higher rate of false positives, as also demonstrated by the t-SNE visualizations of the GNN and reconstructed embeddings in Figure 10. For the NF-CSE-CIC-IDS2018 datasets (both v2 and v3), the performance drop is primarily caused by misclassifications of Infiltration attacks. These involve delivering a malicious payload via email, which then attempts to exploit internal vulnerabilities by scanning the network. Because this behavior closely resembles normal traffic, GraphIDS struggles to reliably classify it as anomalous without prior knowledge of its specific signature, as illustrated in Figures 11 and 12. In contrast, the strong performance on NF-UNSW-NB15-v3 is reflected in the clear separation between benign and attack clusters in Figure 9.

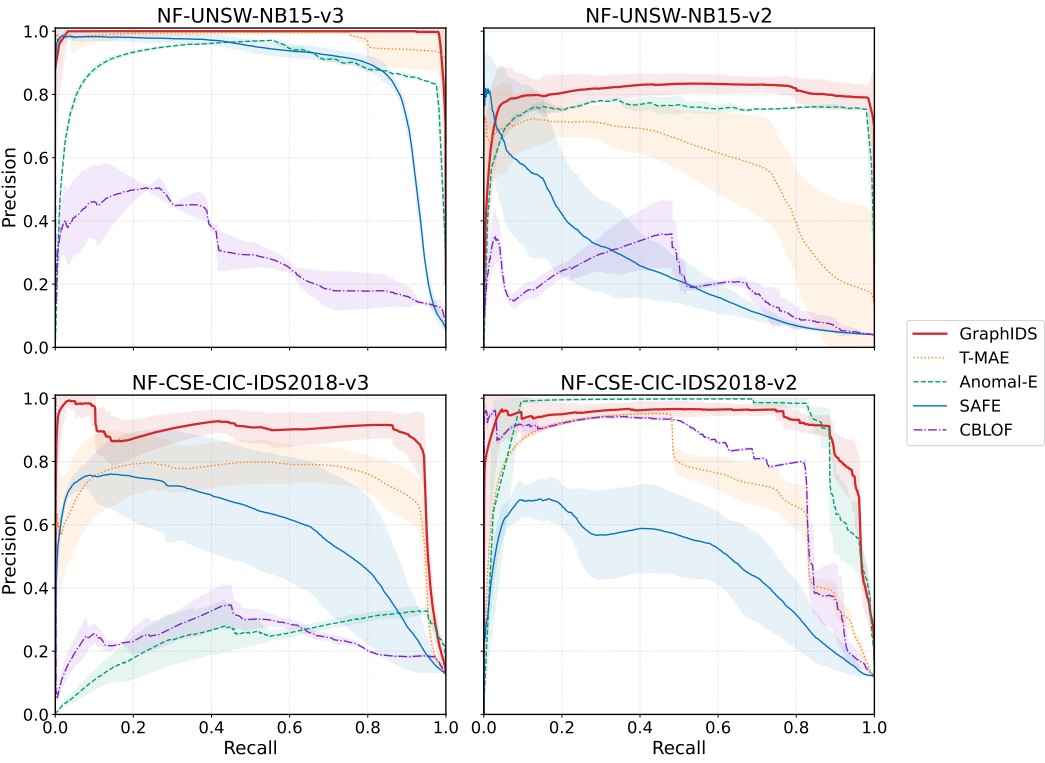

Figure 4: Precision-recall curves for all models on each dataset.

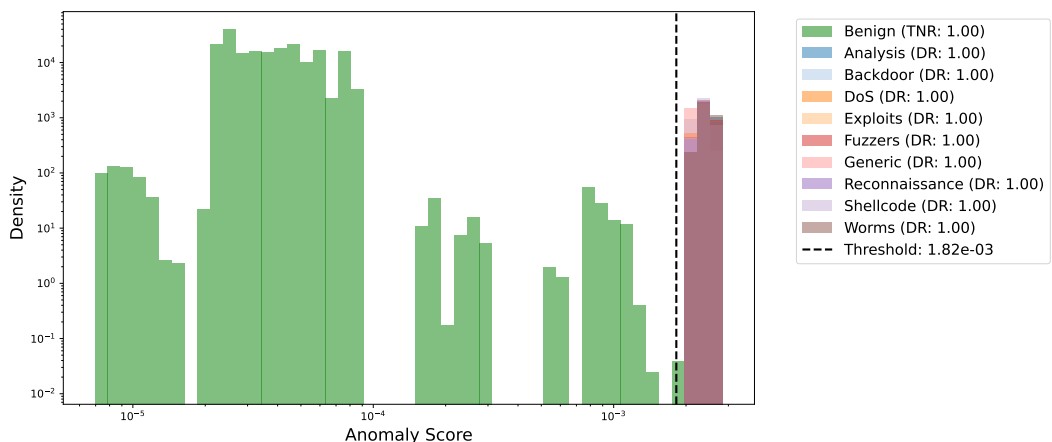

Figure 5: Anomaly score by attack type in NF-UNSW-NB15-v3.

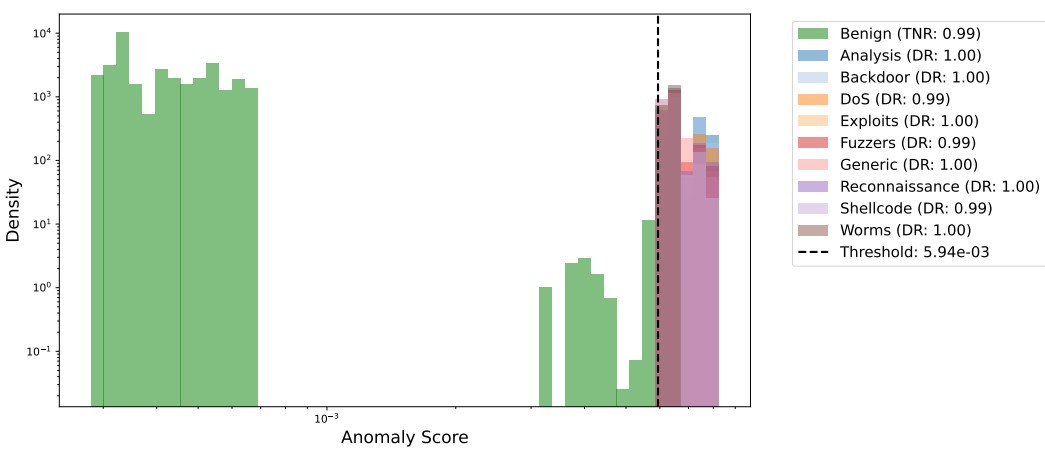

Figure 6: Anomaly score by attack type in NF-UNSW-NB15-v2.

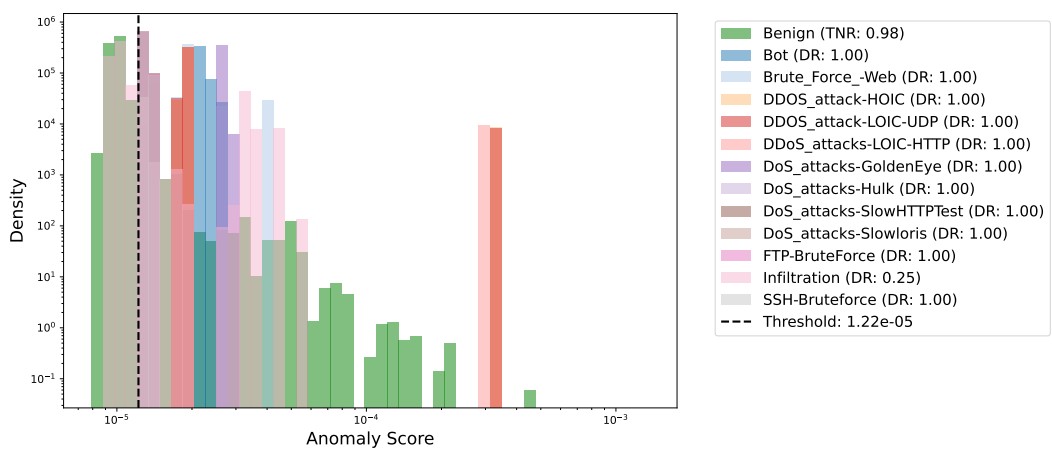

Figure 7: Anomaly score by attack type in NF-CSE-CIC-IDS2018-v3.

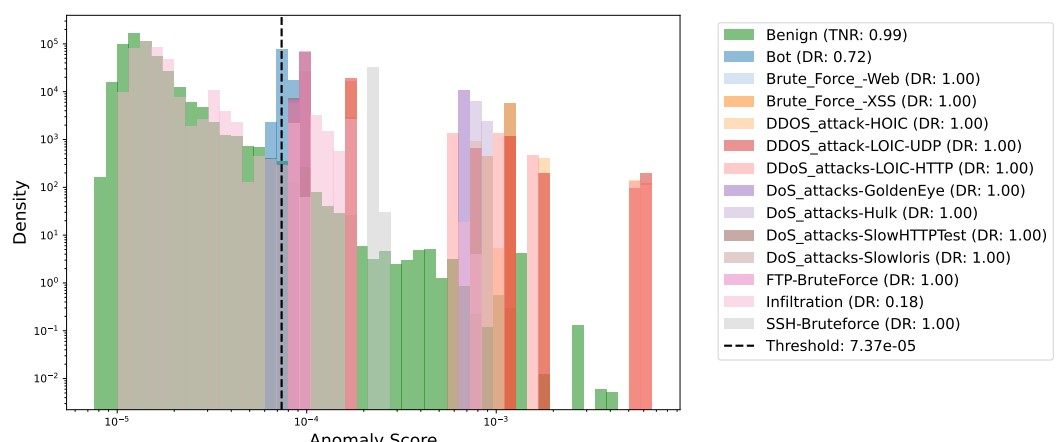

Figure 8: Anomaly score by attack type in NF-CSE-CIC-IDS2018-v2.

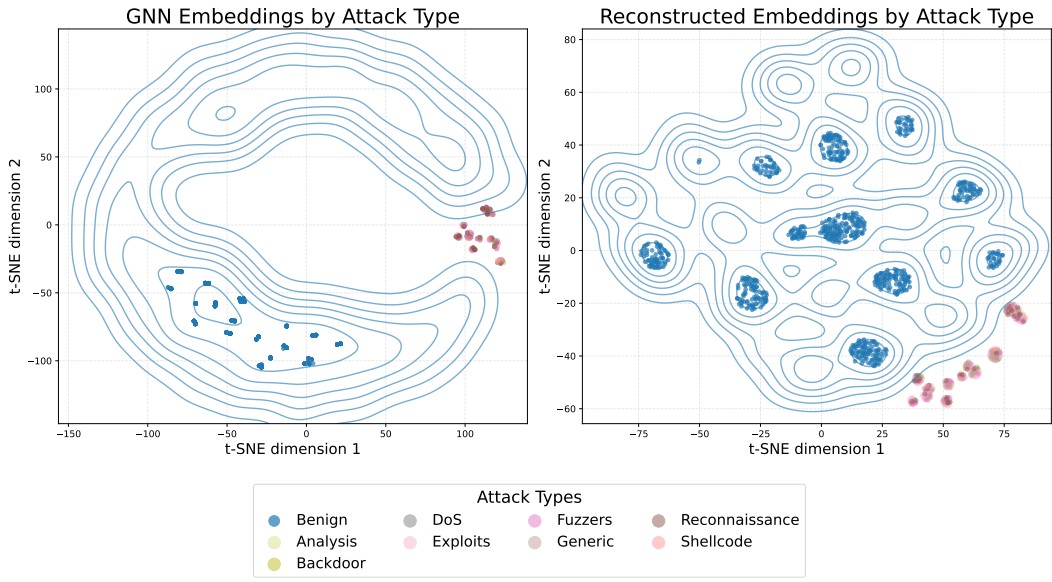

Figure 9: t-SNE visualization of embeddings by attack type in NF-UNSW-NB15-v3, with density contours illustrating the concentration of benign samples.

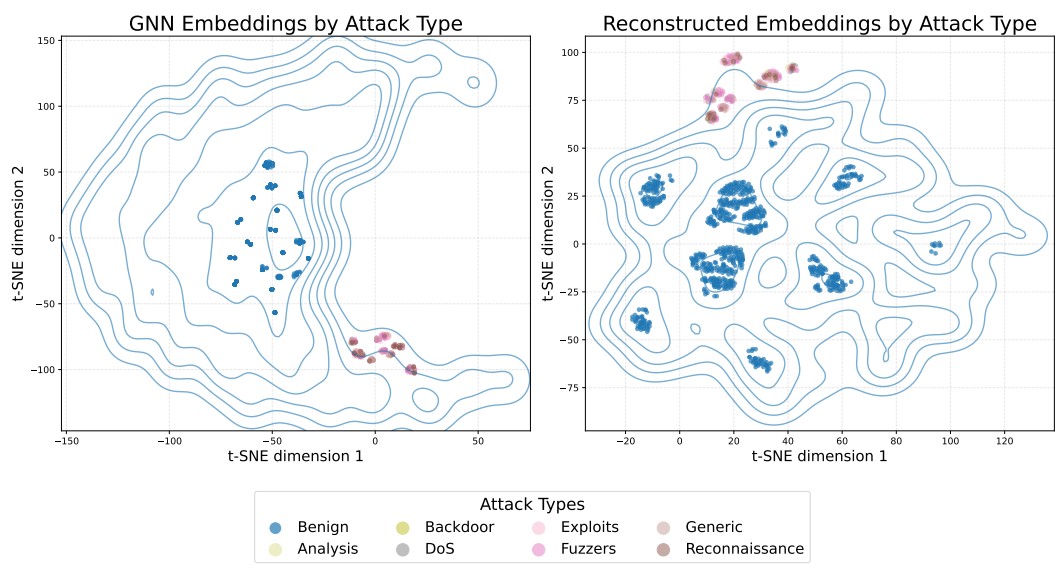

Figure 10: t-SNE visualization of embeddings by attack type in NF-UNSW-NB15-v2.

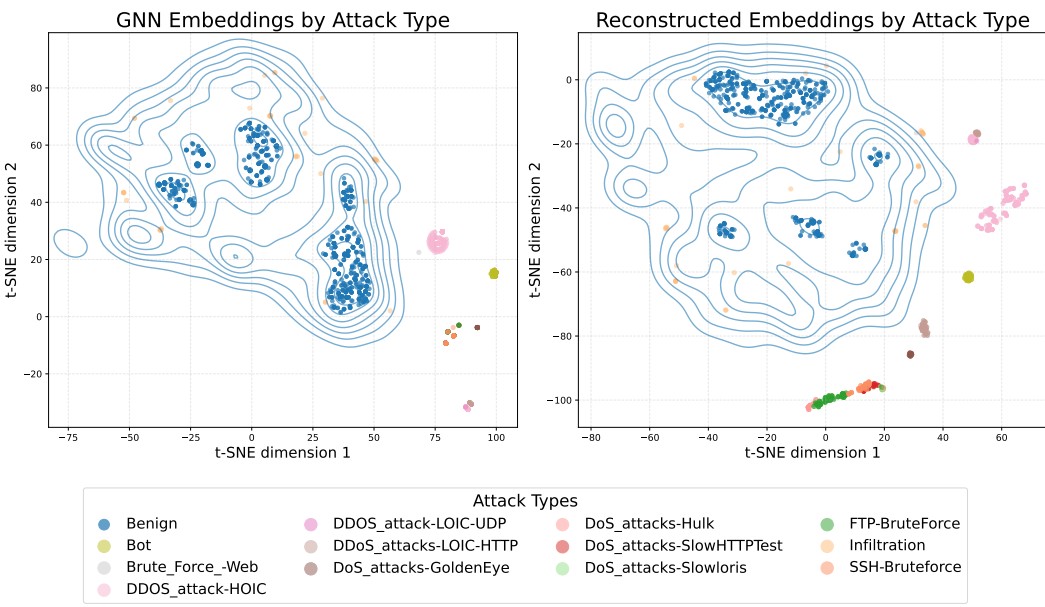

Figure 11: t-SNE visualization of embeddings by attack type in NF-CSE-CIC-IDS2018-v3.

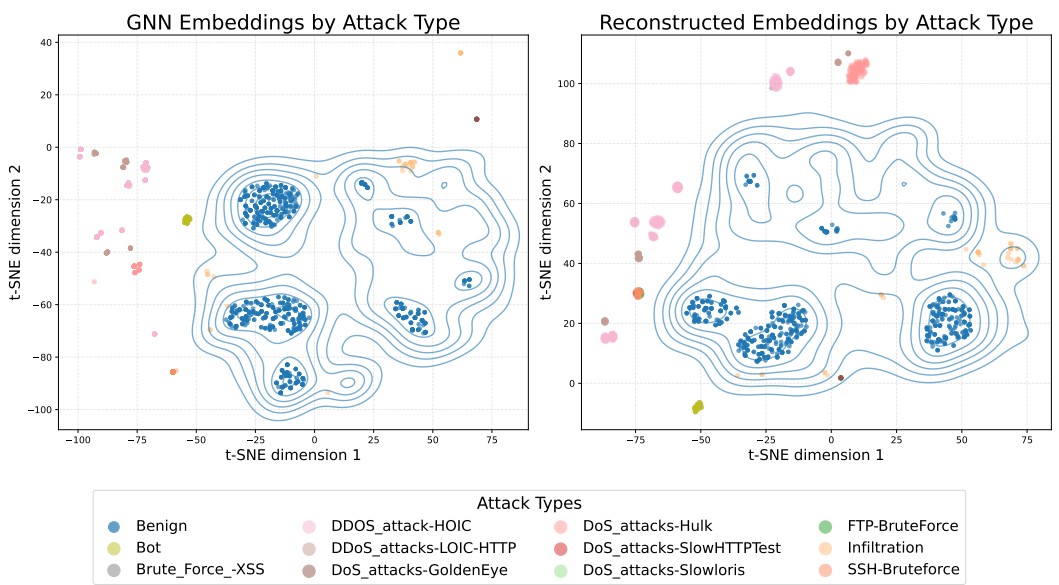

Figure 12: t-SNE visualization of embeddings by attack type in NF-CSE-CIC-IDS2018-v2.

## B.2 Extended Baseline Comparison: Anomal-E and Traditional Methods

In Table 4, we compare the training time and peak GPU memory usage of the evaluated models across datasets. Traditional methods such as CBLOF are excluded from this comparison, as they do not leverage GPU resources.

The results highlight the trade-offs between memory usage, training time, and performance across models. SAFE requires very little GPU memory thanks to its shallow architecture, low input dimensionality, and feature selection, but this comes at the cost of longer training times and substantially lower performance, as shown in Figure 4 and in Table 3. In contrast, Anomal-E's original design, which relies on batch gradient descent and aggregates over the full neighborhood, leads to a considerable memory footprint of up to 30 GB, making it impractical for large graphs. GraphIDS offers a balanced compromise between efficiency and accuracy, using about 1.37 GB of GPU memory and training in 0.87 hours on average. SimpleAE further reduces compute, training in about 0.76 hours with a peak of roughly 428 MB of GPU memory, while remaining competitive in accuracy in the main tables. Finally, GraphIDS's mini-batch strategy allows it to learn normal network behavior from the full graphs of the NF-CSE-CIC-IDS2018 datasets without downsampling, which in our experiments led to improved performance and stability.

Tables 5 and 6 summarize the performance of Anomal-E against the set of anomaly detection algorithms introduced in the original paper. These results show that none of these methods, unlike GraphIDS, is able to maintain a consistent performance across all datasets.

In addition, Tables 7 and 8 report results for traditional anomaly detection algorithms applied directly to raw NetFlow features. All these models show substantially lower performance compared to GraphIDS. Among them, CBLOF achieves the most competitive results on average and is used in the main paper as a representative baseline for traditional methods.

Table 4: Comparison of training time and peak GPU memory usage across models.

| Model | Training Time (h) | Peak Memory (MB) |
|---|---|---|
| SAFE | 3.59 ± 1.34 | 66 |
| Anomal-E | 2.51 ± 1.39 | 29,775 |
| T-MAE | 2.21 ± 2.87 | 9,063 |
| SimpleAE | 0.76 ± 0.65 | 428 |
| GraphIDS | 0.87 ± 0.40 | 1,366 |

Table 5: Performance comparison of different anomaly detection algorithms applied to **Anomal-E** embeddings on the v3 datasets. Results for GraphIDS are included as reference. Bold values indicate statistically significant improvements.

| Model | Metric | NF-UNSW-NB15-v3 | NF-CSE-CIC-IDS2018-v3 |
|---|---|---|---|
| Anomal-E-CBLOF | PR-AUC | 0.7827 ± 0.0840 | 0.2555 ± 0.0383 |
| | Macro F1 | 0.8891 ± 0.1127 | 0.6709 ± 0.0394 |
| Anomal-E-HBOS | PR-AUC | 0.8735 ± 0.0126 | 0.1663 ± 0.0241 |
| | Macro F1 | 0.9458 ± 0.0007 | 0.5329 ± 0.0487 |
| Anomal-E-IF | PR-AUC | 0.7613 ± 0.0530 | 0.1812 ± 0.0218 |
| | Macro F1 | 0.9166 ± 0.0345 | 0.5514 ± 0.0195 |
| Anomal-E-PCA | PR-AUC | 0.9032 ± 0.0041 | 0.1098 ± 0.0165 |
| | Macro F1 | 0.9459 ± 0.0009 | 0.4898 ± 0.0347 |
| GraphIDS (Ours) | PR-AUC | **0.9998 ± 0.0007** | **0.8819 ± 0.0347** |
| | Macro F1 | **0.9961 ± 0.0084** | **0.9447 ± 0.0213** |

Table 6: Performance comparison of different anomaly detection algorithms applied to **Anomal-E** embeddings on the v2 datasets.

| Model | Metric | NF-UNSW-NB15-v2 | NF-CSE-CIC-IDS2018-v2 |
|---|---|---|---|
| Anomal-E-CBLOF | PR-AUC | 0.7175 ± 0.0041 | 0.9287 ± 0.0265 |
| | Macro F1 | 0.9262 ± 0.0008 | 0.9410 ± 0.0161 |
| Anomal-E-HBOS | PR-AUC | 0.7489 ± 0.0074 | 0.9154 ± 0.0181 |
| | Macro F1 | 0.9156 ± 0.0217 | 0.9415 ± 0.0131 |
| Anomal-E-IF | PR-AUC | 0.7438 ± 0.0162 | 0.8847 ± 0.0789 |
| | Macro F1 | 0.9153 ± 0.0216 | 0.9332 ± 0.0302 |
| Anomal-E-PCA | PR-AUC | 0.7133 ± 0.0034 | 0.9178 ± 0.0078 |
| | Macro F1 | 0.9262 ± 0.0008 | 0.9436 ± 0.0076 |
| GraphIDS (Ours) | PR-AUC | **0.8116 ± 0.0367** | 0.9201 ± 0.0238 |
| | Macro F1 | 0.9264 ± 0.0217 | 0.9431 ± 0.0131 |

Table 7: Evaluation of **traditional** anomaly detection algorithms on the v3 datasets.

| Model | Metric | NF-UNSW-NB15-v3 | NF-CSE-CIC-IDS2018-v3 |
|---|---|---|---|
| CBLOF | PR-AUC | 0.3658 ± 0.0634 | 0.2638 ± 0.0263 |
| | Macro F1 | 0.7319 ± 0.0225 | 0.6599 ± 0.0130 |
| HBOS | PR-AUC | 0.2604 ± 0.0021 | 0.1822 ± 0.0011 |
| | Macro F1 | 0.7171 ± 0.0007 | 0.5365 ± 0.0070 |
| IF | PR-AUC | 0.2537 ± 0.0205 | 0.1630 ± 0.0139 |
| | Macro F1 | 0.6822 ± 0.0152 | 0.5330 ± 0.0160 |
| PCA | PR-AUC | 0.4380 ± 0.0038 | 0.1200 ± 0.0003 |
| | Macro F1 | 0.7554 ± 0.0018 | 0.5306 ± 0.0004 |
| GraphIDS (Ours) | PR-AUC | **0.9998 ± 0.0007** | **0.8819 ± 0.0347** |
| | Macro F1 | **0.9961 ± 0.0084** | **0.9447 ± 0.0213** |

Table 8: Evaluation of **traditional** anomaly detection algorithms on the v2 datasets.

| Model | Metric | NF-UNSW-NB15-v2 | NF-CSE-CIC-IDS2018-v2 |
|---|---|---|---|
| CBLOF | PR-AUC | 0.2102 ± 0.0157 | 0.7822 ± 0.0198 |
| | Macro F1 | 0.7046 ± 0.0140 | 0.8889 ± 0.0068 |
| HBOS | PR-AUC | 0.3197 ± 0.0036 | 0.6662 ± 0.0205 |
| | Macro F1 | 0.7032 ± 0.0012 | 0.8578 ± 0.0034 |
| IF | PR-AUC | 0.1914 ± 0.0075 | 0.6124 ± 0.0269 |
| | Macro F1 | 0.6844 ± 0.0071 | 0.8321 ± 0.0099 |
| PCA | PR-AUC | 0.2840 ± 0.0039 | 0.5911 ± 0.0014 |
| | Macro F1 | 0.6975 ± 0.0023 | 0.6128 ± 0.0076 |
| GraphIDS (Ours) | PR-AUC | **0.8116 ± 0.0367** | **0.9201 ± 0.0238** |
| | Macro F1 | **0.9264 ± 0.0217** | **0.9431 ± 0.0131** |

# C   Ablation Studies

In this section, we aim to isolate the individual contributions of specific design choices in GraphIDS. To reduce the computational cost of the ablation study, we conducted experiments over fewer random seeds than those used for the main results. Nevertheless, the setup was sufficient to clearly identify the impact of each component.

## C.1   Effect of Timestamp Features

In this ablation study, we evaluate the impact of including timestamps (`FLOW_START_MILLISECONDS` and `FLOW_END_MILLISECONDS`) among the input features. Our goal is to determine whether they provide useful temporal information or only introduce noise. As shown in Table 9, their inclusion has negligible impact on NF-UNSW-NB15-v3, but clearly degrades the performance on NF-CSE-CIC-IDS2018-v3. This suggests that, overall, excluding timestamps leads to more efficient representations.

Table 9: Effect of including timestamp features on model performance across v3 datasets.

| Model | Metric | NF-UNSW-NB15-v3 | NF-CSE-CIC-IDS2018-v3 |
|---|---|---|---|
| w/ TS | PR-AUC | $0.9991 \pm 0.0011$ | $0.7909 \pm 0.0151$ |
| | Macro F1 | $0.9957 \pm 0.0091$ | $0.9088 \pm 0.0110$ |
| w/o TS | PR-AUC | $0.9989 \pm 0.0017$ | $0.8523 \pm 0.0283$ |
| | Macro F1 | $0.9982 \pm 0.0029$ | $0.9385 \pm 0.0122$ |

## C.2   Effect of Positional Encoding

To investigate whether our model can benefit from sequence modeling, beyond co-occurrence patterns, we temporally ordered the flows in the v3 datasets and added positional encodings to each input window before passing it to the Transformer. We evaluated two variants: sinusoidal and learnable encodings.

For sinusoidal positional encoding, we used the formulation from [27], where the encoding at position $pos$ and dimension $i$ is given by:

$$\mathrm{PE}_{(pos,2i)} = \sin\left(\frac{pos}{10000^{2i/d}}\right), \quad \mathrm{PE}_{(pos,2i+1)} = \cos\left(\frac{pos}{10000^{2i/d}}\right) \tag{3}$$

For learnable positional encoding, we used a parameter matrix $\mathbf{P} \in \mathbb{R}^{L \times d}$, with $L$ as the maximum sequence length and $d$ the embedding dimension. This matrix is optimized along with the rest of the model during training.

The results in Table 10 indicate that positional encodings have little impact on GraphIDS's performance, suggesting that the model primarily learns global co-occurrence patterns rather than temporal dependencies. To reflect this, we omit positional encodings in the main experiments and shuffle the flow order.

Table 10: Effect of positional encoding across datasets.

| Model | Metric | NF-UNSW-NB15-v3 | NF-CSE-CIC-IDS2018-v3 |
|---|---|---|---|
| w/ Learnable Encoding | PR-AUC | $0.9939 \pm 0.0126$ | $0.8572 \pm 0.0284$ |
| | Macro F1 | $0.9873 \pm 0.0263$ | $0.9480 \pm 0.0153$ |
| w/ Sinusoidal Encoding | PR-AUC | $0.9955 \pm 0.0110$ | $0.8537 \pm 0.0421$ |
| | Macro F1 | $0.9904 \pm 0.0214$ | $0.9446 \pm 0.0171$ |
| w/o Positional Encoding | PR-AUC | $0.9955 \pm 0.0126$ | $0.8661 \pm 0.0411$ |
| | Macro F1 | $0.9821 \pm 0.0256$ | $0.9546 \pm 0.0099$ |

## C.3 Effect of GNN Dropout Rate

As shown in Table 11, the dropout rate in E-GraphSAGE has a relevant impact on GraphIDS's overall performance. Higher dropout rates achieved better results on the NF-UNSW-NB15 datasets, suggesting that the GNN is prone to overfitting in smaller network environments. In these cases, stronger regularization helps the model generalize to unseen data. On the NF-CSE-CIC-IDS2018 datasets, introducing a non-zero dropout rate also helped stabilize the learning process, although its impact on final performance was less pronounced.

Table 11: Effect of the GNN dropout rate across datasets.

| Dataset | Metric | 0.0 | 0.25 | 0.5 | 0.6 | 0.7 |
|---|---|---|---|---|---|---|
| NF-UNSW-NB15-v3 | PR-AUC | 0.9773 | 0.9619 | 0.9845 | 0.9998 | 0.9790 |
| | Macro F1 | 1.0000 | 0.9999 | 1.0000 | 1.0000 | 1.0000 |
| NF-CSE-CIC-IDS2018-v3 | PR-AUC | 0.8758 | 0.8645 | 0.8630 | 0.8461 | 0.8621 |
| | Macro F1 | 0.9466 | 0.9270 | 0.9450 | 0.9160 | 0.9219 |
| NF-UNSW-NB15-v2 | PR-AUC | 0.8117 | 0.8154 | 0.8129 | 0.8094 | 0.8301 |
| | Macro F1 | 0.9342 | 0.9303 | 0.9140 | 0.9241 | 0.9285 |
| NF-CSE-CIC-IDS2018-v2 | PR-AUC | 0.8952 | 0.8964 | 0.9005 | 0.8886 | 0.8925 |
| | Macro F1 | 0.9322 | 0.9429 | 0.9411 | 0.9402 | 0.9396 |

## C.4 Effect of Masking Ratio

We found that the masking ratio had a noticeable impact on the model's performance for NF-CSE-CIC-IDS2018-v3. In this case, a masking ratio of 0.15 made the reconstruction task sufficiently challenging for GraphIDS to learn more complex relationships within the flow embeddings. We also noticed that ratios of 0.7 or higher led to gradient explosions and training instability across all datasets.

Table 12: Effect of the attention mask ratio across datasets.

| Dataset | Metric | 0.0 | 0.15 | 0.3 | 0.5 | 0.7 |
|---|---|---|---|---|---|---|
| NF-UNSW-NB15-v3 | PR-AUC | 0.9992 | 1.0000 | 1.0000 | 1.0000 | 1.0000 |
| | Macro F1 | 0.9998 | 0.9999 | 0.9793 | 0.9998 | 0.9998 |
| NF-CSE-CIC-IDS2018-v3 | PR-AUC | 0.8659 | 0.8772 | 0.8691 | 0.8391 | 0.8216 |
| | Macro F1 | 0.9445 | 0.9472 | 0.9485 | 0.9051 | 0.9308 |
| NF-UNSW-NB15-v2 | PR-AUC | 0.8373 | 0.8384 | 0.8384 | 0.8261 | 0.8289 |
| | Macro F1 | 0.9344 | 0.9342 | 0.9344 | 0.9299 | 0.8756 |
| NF-CSE-CIC-IDS2018-v2 | PR-AUC | 0.9115 | 0.9133 | 0.9132 | 0.9132 | 0.9155 |
| | Macro F1 | 0.9390 | 0.9419 | 0.9419 | 0.9398 | 0.9414 |

## C.5 Effect of Neighborhood Size

We explored various neighborhood sizes during the initial development phase, as this choice directly impacts both model performance and computational cost. To rigorously validate our design, we conducted an ablation study comparing 1-hop, 2-hop, and 3-hop variants of GraphIDS under identical experimental conditions.

As shown in Table 13, the 1-hop configuration delivers the most consistent performance. On both NF-UNSW-NB15-v3 and NF-CSE-CIC-IDS2018-v3, the 1-hop baseline outperforms deeper configurations in both PR-AUC and Macro F1. While increasing the receptive field to 2 or 3 hops yields marginal improvements on the v2 versions of these datasets, it does not provide a universal benefit and actually degrades performance on the v3 benchmarks.

Crucially, expanding the neighborhood substantially increases training and inference runtime by up to $3\times$ and, on average, consumes 29% more memory. Although the multi-hop configurations

demonstrate that our architecture can effectively aggregate distant information without suffering from severe over-smoothing, the mixed performance gains are insufficient to justify the added latency. Consequently, we maintain the 1-hop neighborhood as the most effective and efficient default for real-time intrusion detection.

Table 13: Effect of neighborhood size (number of GNN hops) across datasets. Mean (std) over seeds.

| Dataset | Metric | 1-hop | 2-hop | 3-hop |
|---------|--------|-------|-------|-------|
| NF-UNSW-NB15-v3 | PR-AUC | $0.9998 \pm 0.0007$ | $0.9994 \pm 0.0011$ | $0.9977 \pm 0.0031$ |
| | Macro F1 | $0.9961 \pm 0.0084$ | $0.9910 \pm 0.0164$ | $0.9839 \pm 0.0302$ |
| NF-CSE-CIC-IDS2018-v3 | PR-AUC | $0.8819 \pm 0.0347$ | $0.8556 \pm 0.0305$ | $0.8683 \pm 0.0686$ |
| | Macro F1 | $0.9447 \pm 0.0213$ | $0.9352 \pm 0.0094$ | $0.9362 \pm 0.0286$ |
| NF-UNSW-NB15-v2 | PR-AUC | $0.8116 \pm 0.0367$ | $0.8367 \pm 0.0109$ | $0.7766 \pm 0.0814$ |
| | Macro F1 | $0.9264 \pm 0.0217$ | $0.9334 \pm 0.0185$ | $0.9329 \pm 0.0128$ |
| NF-CSE-CIC-IDS2018-v2 | PR-AUC | $0.9201 \pm 0.0238$ | $0.9288 \pm 0.0269$ | $0.9308 \pm 0.0094$ |
| | Macro F1 | $0.9431 \pm 0.0131$ | $0.9475 \pm 0.0211$ | $0.9517 \pm 0.0148$ |

# D   Hyperparameters

Table 14 reports the complete set of optimized hyperparameters used for the GraphIDS model. While these were selected to maximize PR-AUC performance, memory usage can be reduced by decreasing the Transformer's window size, while the GNN remains efficient by randomly sampling subsets of neighboring edges.

Table 14: Hyperparameters for the GraphIDS model across datasets. Dataset names are shortened for formatting.

| Parameter | UNSW-NB15-v3 | NF-CSE-CIC-IDS2018-v3 | UNSW-NB15-v2 | NF-CSE-CIC-IDS2018-v2 |
|-----------|--------------|------------------------|--------------|------------------------|
| *GNN Parameters* | | | | |
| edim_out | 96 | 64 | 72 | 64 |
| nhops | 1 | 1 | 1 | 1 |
| fanout | 32,768 | 32,768 | 32,768 | 32,768 |
| agg_type | mean | mean | mean | mean |
| dropout | 0.6 | 0.5 | 0.75 | 0.5 |
| *Transformer Parameters* | | | | |
| num_layers | 1 | 1 | 1 | 1 |
| num_heads | 4 | 4 | 4 | 4 |
| embed_dim | 48 | 32 | 48 | 32 |
| window_size | 512 | 512 | 512 | 512 |
| mask_ratio | 0.15 | 0.15 | 0.15 | 0.15 |
| dropout | 0.0 | 0.2 | 0.0 | 0.2 |
| *Training Parameters* | | | | |
| learning_rate | $1 \times 10^{-4}$ | $1 \times 10^{-4}$ | $1.1 \times 10^{-5}$ | $7.4 \times 10^{-5}$ |
| gnn_weight_decay | 0.6 | 0.6 | 0.6 | 0.6 |
| ae_weight_decay | 0.04 | 0.04 | 0.046 | 0.011 |
| gnn_batch_size | 16,384 | 16,384 | 32,768 | 16,384 |
| ae_batch_size | 64 | 64 | 64 | 64 |

