# OpenReview forum: "Self-Supervised Learning of Graph Representations for Network Intrusion Detection"
_NeurIPS.cc/2025/Conference — NeurIPS 2025 poster_

### Official Review · Reviewer_dRmA · 2025-06-26

**Clarity:** 3
**Significance:** 2
**Originality:** 3
**Rating:** 4
**Confidence:** 3

**Summary:**

This paper presents GraphIDS, a self-supervised intrusion detection framework that jointly trains a GNN-based encoder and a Transformer-based masked autoencoder to reconstruct benign network flow representations. The method uses NetFlow data to construct a directed graph, where edges correspond to communication flows enriched with statistical features. The model flags anomalous flows by measuring reconstruction error, without requiring labeled attack data. Extensive experiments on four NetFlow-based NIDS datasets demonstrate that GraphIDS outperforms existing baselines by significant margins in both PR-AUC and macro F1, while maintaining fast inference speeds suitable for real-time deployment.

**Questions:**

- How would GraphIDS handle sudden changes in normal network behavior (e.g., flash crowds or reconfigurations)? Is retraining necessary in such cases? How does GraphIDS adapt to evolving benign traffic patterns or changes in network behavior without retraining?
 - Why was a 1-hop neighborhood chosen for the GNN? Would extending to multi-hop improve detection of coordinated attacks? The Transformer masking ratio is fixed at 15%. Was this empirically justified? How does performance vary with different ratios?
 - Since labeled validation data is used to set thresholds, can the method work in fully unsupervised settings without such labels?
 - How resilient is GraphIDS to evasion attempts, where attackers mimic benign traffic patterns to reduce reconstruction error?

**Ethical Concerns:**

["NO or VERY MINOR ethics concerns only"]

**Final Justification:**

The rebuttal has convincingly addressed my primary concerns by providing additional ablation results that justify the 1-hop neighborhood and masking ratio choices, and by clarifying the role of labeled validation data. Although adaptability to sudden benign traffic changes and robustness to evasion attacks remain only partially explored, the proposed joint GNN and Transformer training paradigm, applied for the first time to network intrusion detection, delivers clear and meaningful performance gains over pipeline approaches. Overall, the paper is technically sound, well supported by experiments, and of practical value, and I therefore maintain my Borderline Accept recommendation with the view that its strengths substantially outweigh its remaining limitations.

**Limitations:**

- The model is trained on benign traffic and assumes relatively stable network behavior. In dynamic environments with concept drift or shifting user patterns, detection accuracy may degrade unless retrained or adapted.
  - The effectiveness of GraphIDS depends on meaningful graph structures. In settings with limited topological context (e.g., single-host or encrypted traffic), the model may struggle to learn informative embeddings.
  - Although training is unsupervised, anomaly detection thresholds are selected using a labeled validation set. In practice, such labels may be unavailable, requiring alternative strategies for unsupervised thresholding.
  - Key design decisions, such as the use of 1-hop neighborhoods or fixed attention masking ratio, are not extensively ablated. It remains unclear how these choices affect robustness or generalization.

**Quality:**

3

**Strengths And Weaknesses:**

* Paper Strengths
  - The paper presents an end-to-end architecture that combines GNN and Transformer components in a self-supervised reconstruction task, tailored for network intrusion detection.
  - Extensive experiments across multiple datasets with varied scale and traffic types validate the method’s generalization and robustness, achieving state-of-the-art performance.


* Paper Weaknesses
  - The novelty is moderate, as the method largely combines existing components (E-GraphSAGE, MAE) with limited architectural innovation. The significance of the contribution is limited by the fact that the core idea (reconstruction-based anomaly detection with graph modeling) is well explored in prior work, e.g., GraphMAE and Anomal-E.
  - Though masking strategy is well motivated, more ablation on different masking ratios and their impact on detection performance would be helpful.
  - Inference relies on thresholding over reconstruction errors, which requires some labeled validation data or additional heuristics for deployment. The method still requires labeled validation data for threshold selection, which partially undermines the "unsupervised" claim.

---

> ### Author Rebuttal · Authors · 2025-07-27
>
> We thank the reviewer for the valuable feedback and address each point below.
>
> ---
>
> **1. Handling sudden changes in normal network behavior**
>
> This limitation is acknowledged in Section 5, where we note that GraphIDS assumes relatively stable benign behavior and that abrupt changes may lead to increased false positives or missed detections. We also mention online learning as a promising strategy to address this.
>
> To make this limitation and its implications more prominent, we will emphasize this point more clearly in the revised version and expand the discussion of potential adaptation strategies, such as adaptive thresholding or self-training on high-confidence samples.
>
> We consider this an important direction for future work. A realistic evaluation of the robustness of these systems would require new datasets or experimental setups explicitly designed to reflect dynamic but benign network changes. While both UNSW-NB15 and CSE-CIC-IDS2018 include diverse traffic types and temporal segments, they do not specifically model sudden shifts in benign behavior. Therefore, while some variation is present, they are not ideally suited for studying online adaptation in the face of benign concept drift, and to the best of our knowledge, no existing network intrusion datasets have been designed with this specific goal in mind.
>
> ---
>
> **2. On GNN neighborhood size and masking ratio**
>
> Thank you for raising this important point.
>
> Regarding neighborhood size, we explored various settings during the initial development phase. Preliminary results indicated that increasing the number of hops did not consistently improve performance, while substantially increasing training and inference runtime by up to 3$\times$, and using on average 24% more memory. This not only made extensive hyperparameter tuning impractical but also negatively impacted real-time latency, which is a critical aspect of intrusion detection. Based on these observations, we selected a 1-hop neighborhood as the default, though our implementation supports arbitrary n-hop configurations.
>
> To rigorously validate this design choice, we conducted an ablation study comparing 1-hop, 2-hop, and 3-hop variants of GraphIDS under identical experimental conditions. As shown in the table below, the 1-hop configuration consistently delivers strong and stable performance across all datasets. While the 3-hop variant shows slightly higher scores on NF-CSE-CIC-IDS2018-v3, the differences are within the standard deviation and not statistically significant. In contrast, on other datasets, larger neighborhoods often lead to degraded or less stable performance. This suggests that increasing the receptive field may introduce noise from distant, less relevant nodes, diluting local information. Given the added computational overhead of multi-hop aggregation, these results support our choice of 1-hop as an effective and efficient default.
>
> | Model            | NF-UNSW-NB15-v3         | NF-CSE-CIC-IDS2018-v3   | NF-UNSW-NB15-v2         | NF-CSE-CIC-IDS2018-v2   |
> | ---------------- | ----------------------- | ----------------------- | ----------------------- | ----------------------- |
> | GraphIDS (1-hop) | PR-AUC: 0.9998 ± 0.0007 | PR-AUC: 0.8819 ± 0.0347 | PR-AUC: 0.8116 ± 0.0367 | PR-AUC: 0.9201 ± 0.0238 |
> |                  | F1: 0.9961 ± 0.0084     | F1: 0.9447 ± 0.0213     | F1: 0.9264 ± 0.0217     | F1: 0.9431 ± 0.0131     |
> | GraphIDS (2-hop) | PR-AUC: 0.9980 ± 0.0018 | PR-AUC: 0.8385 ± 0.0528 | PR-AUC: 0.7883 ± 0.0321 | PR-AUC: 0.7539 ± 0.3555 |
> |                  | F1: 0.9935 ± 0.0142     | F1: 0.9370 ± 0.0194     | F1: 0.9147 ± 0.0076     | F1: 0.8371 ± 0.2071     |
> | GraphIDS (3-hop) | PR-AUC: 0.9992 ± 0.0008 | PR-AUC: 0.8969 ± 0.0325 | PR-AUC: 0.8238 ± 0.0521 | PR-AUC: 0.7482 ± 0.3526 |
> |                  | F1: 0.9999 ± 0.0001     | F1: 0.9606 ± 0.0083     | F1: 0.8005 ± 0.2486     | F1: 0.8495 ± 0.2133     |
>
> We will include the full results and discussion in the appendix of the revised manuscript and we thank the reviewer for prompting this investigation.
>
> Regarding the masking ratio, the 15% value was chosen based on ablation experiments detailed in Appendix C.4 (in the supplementary material), where we evaluated ratios of 0%, 15%, 30%, 50%, and 70%. The 15% ratio consistently provided the best balance between performance and stability. To increase the visibility of this study we will clearly reference it in the main text.
>
> ---
>
> **3. Thresholding and use of labeled validation data**
>
> While GraphIDS is trained in a fully unsupervised manner using only benign data, employing a small labeled validation set to select the detection threshold is a standard evaluation practice in unsupervised anomaly detection literature (e.g., Anomal-E, DeepLog), allowing for consistent and rigorous comparison across methods using metrics like F1-score that depend on threshold choice.
>
> We acknowledge this in Section 4.3 (Early Stopping) and consider it reasonable to assume access to a limited set of labeled data (e.g., simulated or manually annotated attacks) solely for threshold calibration in practical settings.
>
> That said, GraphIDS can be deployed in a fully unsupervised setting. Our released code includes a simple thresholding method based on statistical assumptions about reconstruction errors. However, this approach has not been extensively tuned and typically yields lower performance compared to threshold selection using labeled validation data. To assess threshold-independent performance, however, we report PR-AUC as evaluation metric. Developing more robust unsupervised thresholding techniques remains an important direction for future work.
>
> ---
>
> **4. Resilience to evasion attempts**
>
> We agree that resilience to adversarial evasion attacks is an important concern. While comprehensive adversarial evaluation is beyond this paper's scope, GraphIDS has inherent properties that provide some evasion resistance. Attackers would need to simultaneously mimic benign flow patterns at the individual connection level and replicate legitimate network topology relationships. This structural mimicry is significantly more challenging than evading single-feature detectors. That said, we plan to investigate GraphIDS's robustness against such attacks in future work.
>
> ---
> **5. On Architectural Novelty and Practical Significance**
>
>
> While we build upon established components (GNNs, Transformers), our key contribution lies in the novel joint training paradigm specifically designed for network intrusion detection. Unlike GraphMAE (general node classification) or Anomal-E (which does not finetune E-GraphSAGE for the downstream task), we introduce the first unified architecture where GNN and Transformer components are jointly optimized under a shared reconstruction objective for network flow anomaly detection. Our performance improvements represent substantial practical value in network security, where accuracy gains directly translate to fewer missed attacks and reduced false alarms. This is, to our knowledge, the first demonstration that joint GNN-Transformer training significantly outperforms pipeline approaches in network intrusion detection.
>
> ---
>
> **6. On limited topological context and encrypted traffic**
>
> As acknowledged in Section 5, GraphIDS may be less effective in scenarios with limited topological context, such as single-host deployments. However, the UNSW-NB15 dataset represents a relatively small network with few hosts, yet GraphIDS still demonstrates strong performance, suggesting its robustness even in constrained settings. Regarding encrypted traffic, both datasets contain extensive encrypted flows (e.g., HTTPS, SSH). While payloads are inaccessible, NetFlow metadata (which is derived from packet headers) remains available and is unaffected by encryption. This allows GraphIDS to operate effectively on flow-level features, including in encrypted sessions.
>
> ---
>
> We believe these clarifications, along with the new ablation studies we've conducted, further strengthen our contribution to the field.

---

### Official Review · Reviewer_hnjd · 2025-07-01

**Clarity:** 2
**Significance:** 3
**Originality:** 3
**Rating:** 4
**Confidence:** 4

**Summary:**

The paper introduces GraphIDS, an innovative self-supervised Network Intrusion Detection System (NIDS), which combines the strengths of Graph Neural Networks (GNNs) and Transformers into a unified end-to-end framework. GraphIDS leverages masked autoencoding to reconstruct local graph-based embeddings of benign network flows, enabling it to capture both local topology (via GNNs) and global co-occurrence patterns (via Transformers). The joint training of these components under a shared reconstruction objective is notably novel, ensuring flow embeddings are directly optimized for anomaly detection, free from reliance on explicit labeled attack examples or negative samples.

**Questions:**

Please refer to weaknesses. If provided compelling evidence, I may increase my score.

**Ethical Concerns:**

["NO or VERY MINOR ethics concerns only"]

**Final Justification:**

The paper makes a meaningful, practical contribution with excellent results and a coherent end-to-end design. While some robustness and deployment questions remain for future work, the authors’ clarifications and added analyses sufficiently address my core concerns and the other reviewers as well. I recommend the acceptance for this paper.

**Limitations:**

The authors discussed the limitations in this work. Mentioned in Section 5.

**Paper Formatting Concerns:**

No Paper Formatting Concerns.

**Quality:**

2

**Strengths And Weaknesses:**

## Strengths:
- Unified End-to-End Framework:
The innovative integration of GNN and Transformer architectures via masked reconstruction is methodologically novel, ensuring embeddings optimized explicitly for anomaly detection tasks.

- Evaluations:
Clearly outlines dataset selection criteria, preprocessing steps, hyperparameter tuning strategies (including Bayesian optimization), early stopping criteria, and reproducibility considerations, ensuring transparency and repeatability of experiments.

## Weaknesses:

- I found that the absence of a clear discussion on encrypted network flows (e.g., AES-encrypted traffic) in the paper constitutes a significant omission, especially given the prevalence and increasing ubiquity of encrypted communication in modern networks.
The paper’s approach, which heavily depends on NetFlow features and structural relationships within traffic, inherently makes assumptions about feature availability and informativeness:
   - Implicit Assumption:
      The paper implicitly assumes sufficient discriminative power from structural (graph-based) features derived from unencrypted or partially observable traffic. It doesn’t clarify how encryption impacts feature effectiveness or robustness.
   - Potential Overestimation of Performance:
    The impressive empirical results (99.98% PR-AUC, etc.) might significantly deteriorate in realistic deployments where most traffic is encrypted, potentially making the model much less practical or effective.

- I have also concerns on training convergence in P2P Environments:
 Unlike client-server architectures, P2P lacks inherent hierarchical structure (e.g., explicit "client-to-server" patterns), providing fewer distinctive topological cues for distinguishing normal from anomalous flows. And constant peer connections and disconnections produce rapidly evolving graph structures, complicating the learning of stable structural embeddings. Since GraphIDS relies heavily on stable local neighborhoods and global communication patterns for reconstructive learning, highly dynamic or homogeneous networks (such as P2P) might prevent stable embedding convergence, reducing overall detection capability.

- Limited Evaluation Scope:
Empirical validation is restricted primarily to two specific NetFlow datasets. Broader validation against more diverse network environments, encrypted datasets.

- From the perspective of readers, I want to clarify that are minor issues on writing quality and discrepancies.
   - Lack of Clarity in Methodological Details : The joint training process between the GNN and Transformer, while methodologically innovative, from the readers' perspective, we demands precise and explicit descriptions. Ambiguous or vague explanations undermine reproducibility and scientific rigor.

- Limited Discussion on Practical Deployment Challenges :
  Practical concerns (like real-time inference constraints, robustness under realistic network shifts, or encryption considerations) should be explicitly discussed rather than implicitly assumed manageable.

---

> ### Author Rebuttal · Authors · 2025-07-25
>
> We thank the reviewer for their feedback. Below, we respond to each point raised.
>
> ---
>
> **1. Encrypted traffic**
>
> Both the UNSW-NB15 and CSE-CIC-IDS2018 datasets include encrypted traffic. Specifically, this includes extensive HTTPS and SSH flows in CSE-CIC-IDS2018, and SSH and BitTorrent in UNSW-NB15. This has been confirmed through inspection of the raw PCAP files, which are openly available, using tools like Wireshark.
>
> However, as our method operates on NetFlow/IPFIX features, which summarize metadata at the IP and transport layers (e.g., IP addresses, ports, protocols, flow durations, byte counts), the presence of encryption (at higher layers such as TLS) does not impact the feature availability. NetFlow-based approaches, by design, are agnostic to payload encryption. As such, our model is fully compatible with encrypted traffic scenarios, and its effectiveness is not contingent on visibility into packet payloads.
>
> We will clarify in the paper that both datasets include encrypted traffic (e.g., HTTPS, SSH), and that our approach remains compatible due to its reliance on NetFlow metadata.
>
> ---
>
> **2. Training convergence in P2P environments**
>
> While our primary focus is on enterprise-like, common client-server architectures, we note that the UNSW-NB15 dataset used in our experiments includes BitTorrent traffic, which follows a P2P communication model. Therefore, our model has been partially evaluated in the presence of P2P flows.
>
> While these scenarios may not fully reflect highly dynamic or homogeneous P2P networks, they do provide some exposure to non-hierarchical, non-client-server topologies. A more comprehensive analysis of model behavior under fully dynamic P2P environments remains an interesting direction for future work.
>
> We will also add a clarification in the text noting the presence of BitTorrent traffic in the UNSW-NB15 dataset.
>
> ---
>
> **3. Clarity of methodological details**
>
> While the architecture and joint training process are described in the paper and fully available in the released code, we acknowledge that parts of the narrative may benefit from clearer exposition. Due to space constraints, we avoided extensive math or diagrams, but we agree that additional clarity would enhance reproducibility.
>
> In a future revision, we will revise this section to more explicitly describe the masking and reconstruction steps to aid understanding.
>
> ---
>
> **4. Evaluation scope**
>
> NetFlow-based datasets with labeled attacks are limited in number, and the selected datasets are among the most comprehensive, widely used, and representative of real-world intrusion detection settings. Including encrypted traffic as well, we believe these choices are appropriate for evaluating the effectiveness of our proposed approach, which is intended for practical deployment in conventional network environments. However, we do agree that the research community would greatly benefit from the development and release of more up-to-date, comprehensive, and well-structured datasets reflecting evolving network scenarios and attacks.
>
> ---
>
> **5. Deployment considerations**
>
> We acknowledge this limitation in Section 5, noting that accurately assessing real-world detection latency requires a holistic evaluation of the full pipeline, including data collection, aggregation, preprocessing, and inference. While such a deployment-focused study would offer additional insight, it involves extensive engineering effort and infrastructure setup. As such, we consider it outside the scope of this work, which centers on modeling innovations under the common NetFlow-based IDS paradigm.
>
> ---
>
> We believe these clarifications and planned revisions address the reviewer’s concerns and will help improve the final version of the paper.

---

> > ### Comment · Reviewer_hnjd · 2025-08-05
> >
> > Thank you for your detailed response. I appreciate the clarifications you have provided, particularly regarding the presence of encrypted traffic in the datasets and the partial evaluation of P2P scenarios, as well as your willingness to clarify these points in the final version of the paper. Some of my concerns have been addressed by your reply, and I am pleased to see the authors engage thoughtfully with the feedback. I encourage you to further incorporate these clarifications into the final version of the paper, as this will help improve transparency and understanding for future readers. Nevertheless, a number of my key concerns remain, particularly with respect to broader evaluation in more diverse or challenging network environments and additional clarity in methodological exposition. I hope the authors will continue to consider these aspects in future revisions.
> > Thus, I tend to keep my score unchanged.

---

> > > ### Author Response · Authors · 2025-08-05
> > >
> > > We thank the reviewer and we appreciate the opportunity to address the remaining points.
> > >
> > > Regarding the evaluation scope, we respectfully stand by our choice of the UNSW-NB15 and CSE-CIC-IDS2018 datasets. As we have noted, these are the most comprehensive and widely-used benchmarks in the field, offering a robust testbed for network intrusion detection systems. They cover diverse network scales (from small-scale to enterprise), a wide range of modern attacks, and realistic traffic patterns. By evaluating our model on two versions of these datasets, we have also demonstrated its robustness across different feature sets. While creating new benchmarks is certainly an important goal for the research community, we are confident that our current evaluation provides a thorough and realistic assessment of our model's capabilities in common deployment scenarios.
> > >
> > > Regarding the methodological exposition, we appreciate the reviewer's call for greater precision and take this feedback seriously. In the revised manuscript, we will improve the clarity and formalism of this section to ensure it provides scientific rigor and reproducibility. Specifically, we will add the formal definition of the Mean Squared Error loss function and explicitly describe how its gradient is backpropagated through the entire model to jointly update the parameters of both the Transformer and the GNN. Furthermore, we will provide more explicit details on the model's architecture, including the role of the linear projection layers and the structure of the Transformer's encoder-decoder blocks. We believe these additions concretely address the reviewer's concerns about methodological clarity, while the open-source code we provide serves as a final reference for all implementation details.

---

### Official Review · Reviewer_MZJM · 2025-07-02

**Clarity:** 4
**Significance:** 3
**Originality:** 2
**Rating:** 5
**Confidence:** 4

**Summary:**

This paper introduces a self-supervised network intrusion detection method. The method relies on masked autoencoding of graph representations of network activity. Network flow embeddings are extracted using E-GraphSAGE. These embeddings are masked and reconstructed by a Transformer. As with typical autoencoder methods, the reconstruction error for the model trained on nominal data is thresholded for intrusion detection. The authors compare their approach to a few baselines on four intrusion detection datasets. Finally, the authors provide detailed qualitative results, embedding analysis, and ablation in the appendix.

**Questions:**

1. What steps were taken to ensure the baseline comparison is fair? The hyperparameters and their optimization for the proposed method are described, but they are not described for the baseline methods.

2. The Anomal-E baseline is described as using a pre-trained E-GraphSAGE encoder. Is this encoder fine-tuned on this data? If not, was your encoder also not fine-tuned?

3. Since masking has a minor attention has a minor impact on performance and temporal information is not needed, why is a Transformer the right choice? Could the architecture be simpler without it? What contributes to the dramatic increase in performance over approaches like Anomal-E? The T-MAE approach also works very well in some cases, indicating that there is some part of the Transformer or masking that is necessary for strong intrusion detection.

**Ethical Concerns:**

["NO or VERY MINOR ethics concerns only"]

**Final Justification:**

During the rebuttal phase, the authors gave a detailed response answering my questions and addressing my concerns. In particular, the authors resolved my concerns of fairness in the baseline comparisons, clarified the E-GraphSAGE training process, and performed an additional ablation on model architecture. The additional ablation helped resolve some potential design contradictions revealed by other ablations. Overall, all my major concerns have been addressed.

I believe this is a solid conference paper that will be appreciated by the graph machine learning and cybersecurity communities. Therefore, I recommend acceptance.

**Limitations:**

The authors have addressed the limitations but have not discussed negative societal impacts. As this paper focuses on intrusion detection, there are likely no negative societal impacts.

**Quality:**

4

**Strengths And Weaknesses:**

### Strengths

- Strong quantitative performance across multiple intrusion detection datasets and against baseline methods
- Detailed experiments including convincing qualitative and embedding analyses
- Well-written paper with clear and informative figures

### Weaknesses

1. Limited contribution. While the results are strong and are certainly impactful from an intrusion detection perspective, the core method presented seems common for other applications. Masked autoencoders have been used for self-supervised anomaly detection in images. Since the core feature extraction is done through the E-GraphSAGE architecture, the novelty of the approach is limited. However, I do recognize that the proposed approach's performance is strong, so new designs may unnecessarily complicate the task.

2. The results of the ablation seem to contradict parts of the architecture.

    Appendix C.2 shows that positional encoding was not needed and that temporal dependencies are not strong. Why is a Transformer needed then? Would a MLP be enough? Is attention still extremely important? It would be interesting to see another ablation where the autoencoder architecture is changed.

   Along with that, Appendix C.4 shows that the improvement when attention masking is very minor. The task of reconstruction (like with non-masked autoencoder anomaly detection) is more important than reconstructing well with masked attention. Considering these two results, it is unclear what about this architecture is better than E-GraphSAGE with a vanilla MLP autoencoder (similar to Anomal-E). Clarifying this may strengthen the paper.

Please see the questions section for more weaknesses and specific questions relating to the above comments.

### Quality

Overall, this is a high-quality paper. It has strong experimental results, is well-written, and is detailed.

### Clarity

The paper is clear and easy to follow.

### Significance

The paper is likely significant to the field of network intrusion detection and may inspire anomaly detection approaches in other domains, especially those with temporal graphs (like traffic systems).

### Originality

The architecture is novel for network intrusion detection but the ideas presented and used have limited novelty.

---

> ### Author Rebuttal · Authors · 2025-07-26
>
> We thank the reviewer for their thorough review, which not only recognizes the strengths of our work but also provides suggestions for additional proofs that further validate our findings. Below, we respond to each point in detail.
>
> ---
>
> **1. On the fairness of baseline comparisons**
>
> The reviewer raises a valid point. Our goal was to make comparisons as fair as possible under practical constraints:
>
> * T-MAE (our ablation): We conducted dedicated hyperparameter tuning to ensure competitive performance.
> * Anomal-E and traditional anomaly detection models: We tuned the E-GraphSAGE encoder but saw no performance gain, so we retained the original configuration, which was already tuned for the same datasets. We explored and adjusted the hyperparameters of the anomaly detection components using the same ranges as in the original work. These details are available in our code.
> * SAFE: We tuned the hyperparameters of the LOF anomaly detection component, but briefly exploring alternative hyperparameters for the MAE module yielded no performance improvements. Furthermore, as can be observed from the original codebase, the MAE component in SAFE shows poor discrimination ability (e.g., flat ROC curves, AUC=0.5), which limits its contribution regardless of parameter tuning. Given these factors and the computational cost (tuning SAFE can take several days), we believe further tuning would yield marginal gains and not meaningfully affect the overall conclusions. We will clarify this point in the revised manuscript.
>
> ---
>
> **2. On pre-training and fine-tuning of E-GraphSAGE**
>
> In Anomal-E, the E-GraphSAGE encoder is trained using Deep Graph Infomax (DGI) on benign training data, essentially a self-supervised pretext task, so we refer to the encoder as "pre-trained". This step is performed before anomaly detection and the encoder is not fine-tuned afterward, as the original design and the assumption of label unavailability preclude it. Our implementation follows this setup, consistent with the original Anomal-E paper.
>
> In contrast, our method jointly trains E-GraphSAGE together with the Transformer encoder-decoder, optimizing the entire architecture end-to-end without relying on a separate pretext task. This allows the encoder to learn representations that are directly useful for the anomaly detection objective, rather than general-purpose graph embeddings. This difference in training strategy is a central reason for our model’s improved performance.
>
> ---
>
> **3. On the role of the Transformer and performance improvements**
>
> This question addresses a key design choice we considered during the development of GraphIDS. We had explored simpler architectures, and for this rebuttal, we have run a rigorous comparison to provide concrete data on this choice. To do so, we tested a baseline where the Transformer is replaced by a fully connected autoencoder, consisting of a two-layer encoder and a two-layer decoder, using ReLU activations, isolating the architectural benefit on top of our end-to-end reconstruction framework. To ensure a fair comparison, we explored different bottleneck dimensions and hyperparameters.
>
> | Model                 | NF-UNSW-NB15-v3         | NF-CSE-CIC-IDS2018-v3   | NF-UNSW-NB15-v2         | NF-CSE-CIC-IDS2018-v2   |
> | ---------------------| ------------------------| ------------------------| ------------------------| ------------------------|
> | **SimpleAutoencoder**| PR-AUC: 0.9996 ± 0.0007  | PR-AUC: 0.8458 ± 0.0498 | PR-AUC: 0.7864 ± 0.0608 | PR-AUC: 0.9310 ± 0.0126 |
> |                      | F1: 0.9838 ± 0.0321      | F1: 0.9223 ± 0.0201     | F1: 0.8680 ± 0.1579     | F1: 0.9450 ± 0.0092     |
> | **GraphIDS**         | PR-AUC: 0.9998 ± 0.0007  | PR-AUC: 0.8819 ± 0.0347 | PR-AUC: 0.8116 ± 0.0367 | PR-AUC: 0.9201 ± 0.0238 |
> |                      | F1: 0.9961 ± 0.0084      | F1: 0.9447 ± 0.0213     | F1: 0.9264 ± 0.0217     | F1: 0.9431 ± 0.0131     |
>
>
> The results confirm that the autoencoding task itself provides the most significant performance increase, as evidenced by the strong SimpleAutoencoder baseline.
>
> However, the Transformer still provides clear improvements in performance and stability. GraphIDS consistently achieves higher Macro F1-scores, which we attribute to self-attention's ability to capture global co-occurrence patterns between flows. Importantly, we gain this advantage while keeping the model lightweight, using only a single-layer Transformer encoder and decoder. While we did not focus on maximizing its efficiency, it is also possible to reduce the window size to balance performance with resource usage.
>
> We will include these new results in the appendix and expand the architectural discussion in the revised paper.
>
> ---
>
> We thank the reviewer for raising these technical points, and we believe these additions further support the robustness and practical value of our approach.

---

> > ### Comment · Reviewer_MZJM · 2025-08-04
> >
> > Thank you for your detailed response. The majority of my comments and questions have been addressed. I especially appreciate the added analysis on the role of the Transformer. It is interesting to see that the SimpleAutoencoder performed so well. On some datasets, the performance was the same, or even lower, when using the Transformer. I wonder why this result seems dataset-dependent. Is there some aspect of these datasets that makes the ability to capture global co-occurrence patterns between flows unnecessary? Hopefully, an explanation for this can be added along with the new results in the appendix.
> >
> > Overall, I still feel positive about this paper. I will consider the authors' responses to my comments and other reviewers' comments when giving my final rating of the paper.

---

> > > ### Author Response · Authors · 2025-08-04
> > >
> > > Thank you for your follow-up comment. You are correct in noticing that while the Transformer outperforms the SimpleAutoencoder by a wider margin on two datasets, the performance difference is smaller on the others. This disparity is indeed due to fundamental differences between the datasets.
> > >
> > > As noted in Section 4.1, UNSW-NB15 and CSE-CIC-IDS2018 represent radically different network environments and attacks. Furthermore, the v2 and v3 versions vary in their feature sets and extraction processes. This means some attacks can be more easily identified through specific features present in one dataset version but are more subtle in another. This can be noticed by the significantly higher performance on the NF-UNSW-NB15-v3 compared to the v2, and this highlights the point we make in Section 4.6 about the importance of carefully engineering NetFlow features for any specific deployment.
> > >
> > > This explains why SimpleAutoencoder and GraphIDS can perform similarly when the features themselves already provide clear signals for detection. However, our goal is to design a system that is robust not only with perfectly engineered features but also in scenarios with more subtle attack patterns that require understanding the broader context. This is where the Transformer's architecture shows its advantages. Its self-attention mechanism is better at capturing complex, global relationships between flows, making it more consistent and reliable across the diverse scenarios we tested and thus less dependent on extensive feature engineering.
> > >
> > > For resource-constrained environments, the SimpleAutoencoder remains a valid and lightweight alternative. Its strong performance confirms that our core contribution, the end-to-end training of graph representations for this task, is the main driver of the performance gains.
> > >
> > > We will add this explanation along with the new results, and we thank the reviewer for their valuable suggestion.

---

### Note · Authors · 2025-08-12

We would like to thank the program chairs for the opportunity to provide these final remarks and the reviewers for their time and effort.

The review process has been very constructive and we have conducted two additional ablation studies to address the reviewers' questions. A comparison with a simpler autoencoder baseline clarifies the contribution of our approach, showing that the end-to-end training is the main driver of the performance gains by optimizing graph representations specifically for the reconstruction task, while demonstrating that the Transformer improves performance and stability by capturing global co-occurrence patterns among flows (addressing Reviewer MZJM’s questions). A second study, in response to Reviewer dRmA, empirically validates our 1-hop architecture as the most effective and efficient choice. We also addressed Reviewer hnjd’s concerns by confirming that our evaluation on the UNSW-NB15 and CSE-CIC-IDS2018 datasets inherently covers extensive encrypted traffic and provides exposure to P2P traffic as well, representing a realistic, modern network environment. Finally, we committed to improving the clarity of the methodological details in Section 3.

With these new results and additional clarifications in the final manuscript, we are confident in the robustness of our contribution, and we thank the reviewers once again for their feedback.

---

### Decision · Program_Chairs · 2025-09-17

**Decision:**

Accept (poster)

**Comment:**

This paper introduces GraphIDS, a self-supervised framework for network intrusion detection. Unlike prior approaches that decouple embedding and detection, GraphIDS employs a masked autoencoder to learn normal communication patterns by embedding each flow with its local graph context via an inductive GNN, and reconstructing these embeddings through a Transformer encoder–decoder to capture global co-occurrence structures.  The approach demonstrates strong empirical results on multiple NetFlow benchmarks, achieving up to 99.98% PR-AUC and 99.61% macro F1, significantly outperforming baselines.
All the reviewers gave this paper a positive score. During the rebuttal, the authors adequately address the reviewers' concern.
Overall, the work is well-motivated, technically sound, and presents a compelling end-to-end design that advances graph-based intrusion detection. I recommend acceptance.